# Towards better understanding and better generalization of few-shot classification in histology images with contrastive learning

**Jiawei Yang**[1,2,*,†], **Hanbo Chen**[1,*], **Jiangpeng Yan**[1,3,†], **Xiaoyu Chen**[1,4,†], **Jianhua Yao** [1,*]

[1]Tencent AI Lab, [2]UC Los Angeles, [3]Tsinghua University, [4]Xiamen University

## Abstract

Few-shot learning is an established topic in natural images for years, but few work is attended to histology images, which is of high clinical value since well-labeled datasets and rare abnormal samples are expensive to collect. Here, we facilitate the study of few-shot learning in histology images by setting up three cross-domain tasks that simulate real clinics problems. To enable label-efficient learning and better generalizability, we propose to incorporate *contrastive learning* (CL) with *latent augmentation* (LA) to build a few-shot system. CL learns useful representations without manual labels, while LA transfers semantic variations of the base dataset in an unsupervised way. These two components fully exploit unlabeled training data and can scale gracefully to other label-hungry problems. In experiments, we find i) models learned by CL generalize better than supervised learning for histology images in unseen classes, and ii) LA brings consistent gains over baselines. Prior studies of self-supervised learning mainly focus on ImageNet-like images, which only present a dominant object in their centers. Recent attention has been paid to images with multi-objects and multi-textures (Chen & Li, 2020). Histology images are a natural choice for such a study. We show the superiority of CL over supervised learning in terms of generalization for such data and provide our empirical understanding for this observation. The findings in this work could contribute to understanding how the model generalizes in the context of both representation learning and histological image analysis. Code is available at https://github.com/TencentAILabHealthcare/Few-shot-WSI.

## 1 Introduction

Histological images provide crucial phenotypical and diagnostic information for disease assessment and prognosis (Srinidhi et al., 2020). Thus, computer-aided histological image classification systems are highly demanded but expensive to build due to the scarcity of well-annotated data. Besides, histological images diversify in many aspects, including acquisition protocols, body sites and tissue types. Such heavy domain shifts and variations pose more challenging data-hungry issues. How to train robust models with limited annotated samples becomes the key of general diagnosis systems.

We sort for *few-shot learning* (FSL) to tackle the aforementioned issues. Recent works have demonstrated FSL's success in natural images, but it is much unexplored in histological image analysis. Here, we facilitate the study of FSL and generalized FSL (GFSL) for histology images by setting up 3 cross-domain tasks where there exist near-, middle- and out-domain shifts from base class to novel class. In addition, we investigate the impact of homogeneous and heterogeneous shot selection problem, *i.e.*, few-shot samples come from the same whole slide image (WSI) or different ones.

To enable label-efficient learning and better generalizability, we propose to incorporate *contrastive learning* (CL) with *latent augmentation* (LA) to build a few-shot system. Concretely, CL learns a meaningful encoder during pre-training, while LA inherits knowledge from "unlabeled" base datasets by transferring semantic variations in latent space. These two components fully exploit a base dataset by using its legacy: the learned model weights, and the captured latent variations.

---

*: Equal contribution. †: Work done during intern at Tencent AI Lab. *: Corresponding author.

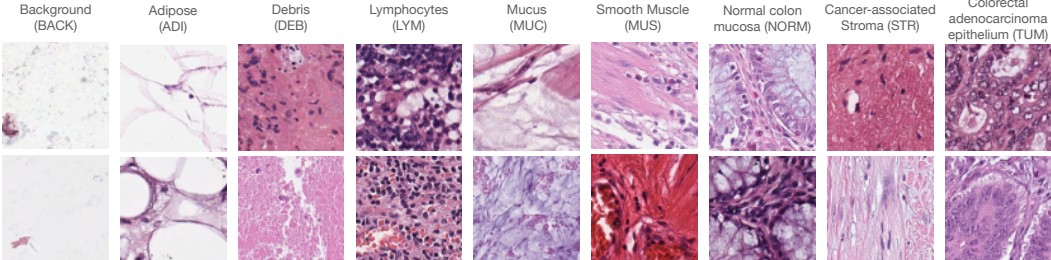

Figure 1: **Example images from NCT (Kather et al., 2018).** Each column contains two samples from the same class (column name). More example images can be found in Appendix A.1.

Also to our surprise, we show the generalization gap between state-of-the-art CL models and the supervised models is larger in histological images than that in natural images. Previously studies of CL mainly focus on *iconic* natural images where only a dominant object occupies image centers, while images like histological ones, where multiple small objects (e.g. cells, nucleus) and various textures (e.g. muscle, mucus) are densely presented, are much unexplored. We also aim to fill the gap of studying CL for non-iconic, multi-object and multi-texture histological images, and empirically explain why the large generalization gap between CL models and supervised ones exists for them.

To summarize, our key findings and contributions are:

- We, as one of early works, study FSL in histological data with domain-specific problems.

- We propose a simple label-efficient method for few-shot learning, which incorporates contrastive learning and latent augmentation to fully exploit training data in an unsupervised way. Extensive experiments confirm their consistent gains and improved generalizability.

- We show that, at slight odds with findings in iconic natural images, CL learned models generalize better than the supervised counterparts for histology images by a large margin. We analyze and provide our empirical explanations for this observation, which we believe could contribute to understanding how model generalizes to novel samples in the context of representation learning and histology image analysis.

## 2 PRELIMINARIES AND PROBLEM FORMULATION

**Whole-Slide Image (WSI).** Whole-slide images are digital scans of histology tissue slides collected by biopsy or surgery. Given micron-size pixels and centimeter-size slides, a WSI is usually of gigapixel size and thus have to be divided into hundreds or thousands of small "patches" for computational analysis. A patch represents the basic unit in patch-level classification problem. As tissues context and staining quality could vary across WSIs, the extracted patches' styles are confined to their source WSIs, leading to inter-WSI domain shift. Besides, unlike iconic natural images which only present a dominant object in their centers, histological patches contain multiple small objects and multiple texture-like tissues (see Figure 1), making the process different from the major recognition systems which only need to focus on dominant objects.

**Few-shot Learning (FSL).** Few-shot classification is to learn from a large "base" dataset, and afterwards generalize to unseen classes with only limited labeled data. Formally, a base dataset is defined as $\mathcal{D}_{base} = \{(\mathbf{x}_i, y_i)\}_{i=1}^{N_{base}} \subset \mathcal{X}_{base} \times \mathcal{Y}_{base}$, where $\mathcal{X}_{base}$ is a sample set and $\mathcal{Y}_{base}$ is their label space. Novel dataset $\mathcal{D}_{novel} = \{(\mathbf{x}_i, y_i)\}_{i=1}^{N_{novel}}$ has a *disjoint* label space, i.e. $\mathcal{Y}_{base} \cap \mathcal{Y}_{novel} = \emptyset$, where $\mathcal{Y}_{novel}$ is the novel label space. A few-shot learner is trained on $\mathcal{D}_{base}$ and evaluated on a series of *meta-tasks* sampled from $\mathcal{D}_{novel}$. Such task is defined as $\mathcal{T} = \{(\mathcal{S}_i, \mathcal{Q}_i)\}_{i=1}^{I}$, where $\mathcal{S} = \{(\mathbf{x}_i, y_i)\}_{i=1}^{NK} \sim \mathcal{D}_{novel}$ is a small training set, called *support* set, and $\mathcal{Q} = \{\mathbf{x}_i\}_{i=1}^{NQ} \sim \mathcal{X}_{novel}$ is another small test set, called *query* set, and $I$ is the number of tasks. This formulation is termed as $N$-way $K$-shot ($Q$-query) task, since $N$ classes are sampled from $\mathcal{Y}_{novel}$, each with $K$ labeled samples for training and $Q$ unlabeled samples for testing. Usually, $K$ is less than $Q$, e.g. $K = 1, 5$, and $Q = 15$. The evaluation stage is often referred to as meta-testing stage.

**Generalized few-shot learning (GFSL).** Unlike in FSL, GFSL samples meta-tasks from a joint dataset $\mathcal{D}_{joint} = \mathcal{D}_{base} \cup \mathcal{D}_{novel}$, with a joint label space $\mathcal{Y}_{joint} = \mathcal{Y}_{base} \cup \mathcal{Y}_{novel}$. Now, support sets and query sets contain both seen base classes and unseen novel classes.

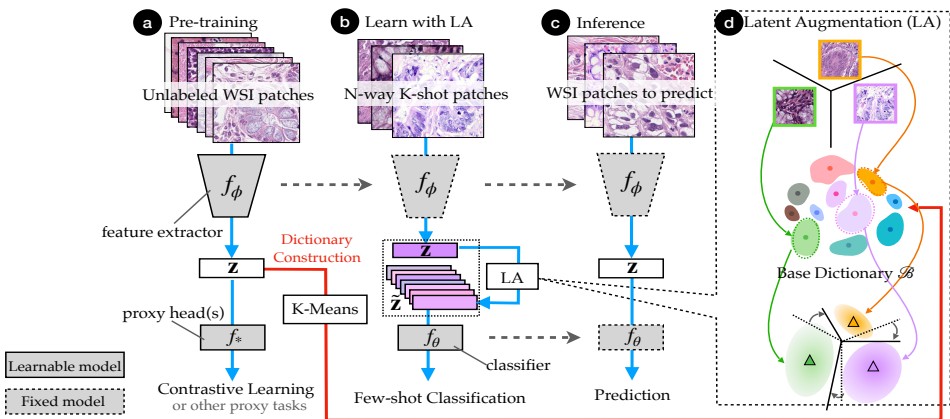

Figure 2: **Overview.** With pre-trained feature extractor (a), N-way K-shot classifiers are learnt (b) based on LA (d) to classify WSI patches (c). Given a novel representation $\mathbf{z}$, LA generates its new features from the most likely variation in the base dictionary, so few-shot novel samples can be proliferated in a reasonable way, and the decision boundary could therefore be improved.

## 3 METHODS

Consider a few-shot classifier as $f = f_\theta \circ f_\phi$, where $f_\phi$ is an embedding function, *i.e.*, a feature extractor that maps a high-dimensional input image $\mathbf{x} \in \mathbb{R}^{3HW}$ into a low-dimensional latent space $\mathbb{R}^d$, and $f_\theta$ is a classifier trained on support set $\mathcal{S}$ and predicts results for query set $\mathcal{Q}$. $\phi$ and $\theta$ are their corresponding parameters. Our method consists of two phases — a) pre-training $f_\phi$ on base datasets and b) training $f_\theta$ on support sets with latent augmentation during meta-testing stage. Figure 2 shows the overview of our methods. We elaborate them in the followings.

### 3.1 PRE-TRAINING

Current paradigms in FSL for training $f_\phi$ lie in two folds: i) meta-training, also known as *episodic* training, where base datasets are divided into various episodic $N$-way $K$-shot meta-tasks that simulate meta-learning; and ii) standard training, which does standard fully supervised classification pre-training without splitting data. The former one emphasizes the idea of meta-learning for fast adaption (Schmidhuber, 1987; Finn et al., 2017), while the latter one attributes the success of FSL to feature reuse (Raghu et al., 2019) or good representations (Chen et al., 2019; Tian et al., 2020). We follow the latter one and we believe better learned encoders lead to stronger generalizability.

**Fully-supervised pre-training (FSP).** Given a base dataset, we jointly train a feature extractor $f_\phi$ and a proxy classifier $f_\psi$ using the standard cross-entropy loss. Once pre-trained, only $f_\phi$ is kept and *fixed* for downstream tasks. We term the embedding functions learned by FSP as $f_\phi^{FSP}$.

**Contrastive-learning pre-training (CLP).** Self-supervised learning methods alleviate the need for data annotation. Here we focus on a contrastive learning method – MoCo-v3 (Chen et al., 2021b), which currently holds the state-of-the-art performance. It consists of three components: a feature extractor (backbone) $f_\phi$, a projection head $f_g$ and a prediction head $f_q$. Given an unlabeled base training dataset $\mathcal{D}_{base}^u = \{\mathbf{x}_i\}_{i=1}^{N_{base}}$, the model learns to minimize the contrastive loss function w.r.t. unlabeled batch data:

$$\phi^*, g^*, q^* = \underset{\phi, g, q}{\arg\min}\, \mathbb{E}_{\mathbf{x}, \mathbf{x}' \overset{t}{\sim} \mathcal{D}_{base}^u} \left[ \mathcal{L}_{CLP} \left( f_q \circ f_g \circ f_\phi(\mathbf{x}), f_{\tilde{g}} \circ f_{\tilde{\phi}}(\mathbf{x}'); \phi, g, q \right) \right], \tag{1}$$

where $\mathcal{L}_{CLP}$ is a contrastive loss function; $\mathbf{x}, \mathbf{x}'$ are two views of the same images obtained by applying random data augmentation $t$; $\tilde{\phi}$ and $\tilde{g}$ denote the momentum updated copies of $\phi$ and $g$. In short, contrastive learning aims to maximize the similarity between positive pairs (two augmented views of a same image), while minimizing the similarity between negative pairs (two different images). We leave more detailed descriptions of MoCo-v3 and $\mathcal{L}_{CLP}$ to Appendix B. Once CLP is done, two auxiliary heads, $f_g$ and $f_q$, are removed, while $f_\phi$ is kept and *fixed*, termed as $f_\phi^{CLP}$.

### 3.2 LATENT AUGMENTATION

The pre-trained feature extractor $f_\phi$ only transfers parts of available knowledge in base datasets by reusing the learned weights. The more transferable knowledge is inherent in *data representations*. It

is reasonable to assume that base classes and novel classes share similar modes of variations (Wang et al., 2018) since they are all histology-related. Such inductive biases allow us to transfer variations from seen tissues or styles to unseen ones. Here we propose to transfer the representation variations in a simple *unsupervised* way. Below, we first introduce *latent augmentation* and then discuss our motivations and intuitions about it.

**Base dictionary and Latent augmentation (LA).** Our goal is to fully exploit training data. This is done not only by reusing pre-trained model weights $f_\phi$, but also by transferring possible semantic shifts of clustered representations. Given an unlabeled base dataset, we perform K-Means on the representations $\mathbf{z} = f_\phi(\mathbf{x})$ to obtain $C$ clusters (Figure 2-(a), red arrows). The *base dictionary* is constructed as $\mathcal{B} = \{(\mathbf{c}_i, \Sigma_i)\}_{i=1}^C$, where $\mathbf{c}_i$ is the $i$-th cluster prototype (*i.e.*, mean representation) and $\Sigma_i$ denotes its intra-cluster covariance matrix. Roughly, $\mathcal{B}$ captures how the pre-trained $f_\phi$ thinks samples from base dataset would vary in latent space conditioned on cluster $i$, *e.g.*, using a multivariate Gaussian $\mathcal{N}(\mathbf{c}_i, \Sigma_i)$. Given the base dictionary $\mathcal{B}$, during meta-testing stage, LA uses original representations $\mathbf{z}$ to query the most likely variations from $\mathcal{B}$, followed by additive augmentation $\tilde{\mathbf{z}} = \mathbf{z} + \boldsymbol{\delta}$ (Figure 2-(b,d)). This is done by sampling $\boldsymbol{\delta} \sim \mathcal{N}(\mathbf{0}, \Sigma_{i^*})$ where $i^*$ is selected by finding the maximum cosine similarity between $\mathbf{z}$ and $\mathbf{c}_i$. The classifier $f_\theta$ is then trained on both the original representations $\mathbf{z}$ and the augmented representations $\tilde{\mathbf{z}}$ (Figure 2-(c)).

### 3.2.1 Intuitions and motivations on latent augmentation

**Why transferring variations works.** LA aims at transferring the knowledge of variations. Such knowledge brings *semantic* diversity from base classes to novel classes. For example, tumorous cells are mutated from normal cells; when given limited tumorous samples, LA may replicate how normal cells alter under this mutation using the captured variations in base dictionaries. This closely resembles how a pathologist builds his/her knowledge on unseen phenotypes from seen phenotypes. From the view of under-representative learning (Yin et al., 2019), replicating latent variations encourages under-represented distributions to be closer to regular ones. From the view of low-data learning, distribution of few samples is not well calibrated (Yang et al., 2021), so using established distributions in base class to calibrate untrustworthy novel class may help. Besides, LA can be seen as a consistency regularization technique. Enforcing the classifiers' predictions to be consistent across different perturbation is known to help in low-data regime (Bachman et al., 2014; Berthelot et al., 2019; Sohn et al., 2020). In fact, LA is a stronger alternative against data augmentation (DA), as we later show in §4.3 that LA outperforms DA by a large margin and can cover the role played by DA.

**Why linear additive augmentation is meaningful.** When well trained, deep networks are hypothesized to be good at linearizing deep features (Bengio et al., 2013; Upchurch et al., 2017). This gives the rationality behind *linearly* inter/extrapolating features, *i.e.*, using "add" operation to generate new features. Recently, Cheung & Yeung (2020) study the universal label-preserving additive augmentations in latent space that can be used in different data modalities, showing the effectiveness of simple linearly transforming latents.

**Why base dictionary construction is warranted in both FSP and CLP.** FSP uses classification task as a proxy task to learn useful encoders $f_\phi$. During optimization, features are incentivized to maximize their dot-product similarity with class weights in $f_\psi$, thus forming meaningful metric space. In the regard of CLP, the contrastive loss, a form of metric-based loss, brings similar features closer while spreading dissimilar representations farther, which also results in an informative metric space. Thus, feature distance in the representation space of both FSP and CLP is meaningful, justifying the use of unsupervised clustering method to construct base dictionary.

## 4 Experiments

### 4.1 Setup

**Datasets.** Since tissues vary across body sites, we use three public histology datasets from different body sites to construct three tasks with different degrees of domain shift. They are: NCT-CRC-HE-100K collected from colon site (Kather et al., 2018), LC25000 collected from lung and colon sites (Borkowski et al., 2019), and PAIP19 collected from liver site (Kim et al., 2021); we term them as NCT, LC-25K and PAIP respectively. NCT consists of 9 classes with 100k non-overlapping patches in total, each of size $224 \times 224$. LC-25K has 5 classes with 5,000 patches in each class; each patch is of size $768 \times 768$. PAIP is composed of 50 WSIs, each of size about $45k \times 45k$ with 3 mask annotated classes. For LC-25K, all patches are resized to $224 \times 224$. For PAIP, the foreground tissues

are cropped into 75k patches of size $224 \times 224$ with the same pixel resolution as NCT and labels are assigned by majority voting. More details are in Appendix A. When novel and base classes are from different organs, we view the novel classes as **out-domain** classes. When from the same organ, we consider them as **near-domain** classes only when the data is collected from the same source. Otherwise, they are considered as **middle-domain** classes due to difference in imaging protocols.

**Task i) Near-domain task (to study GFSL).** We randomly split NCT by 80%/20% to construct a training set (80k images) and a test set (20k images). Then, we do leave-one-class-out-as-novel-class to the training set to construct 9 base datasets and use the test set as $\mathcal{D}_{joint}$ for evaluation. This results in 9 sub-tasks, each of which has one class regarded as novel class and samples belonging to it are excluded from pre-training datasets.

**Task ii) Mixture-domain task (to study FSL).** We use the entire training set (80k images) of NCT as $\mathcal{D}_{base}$, and use LC-25K as $\mathcal{D}_{novel}$ for evaluation. Two of five classes in LC-25K are colon-related (middle-domain novel classes) and the remaining three are lung-related (out-domain novel classes).

**Task iii) Out-domain task (to study FSL).** Similar to mixture-domain task, we use the NCT training set as $\mathcal{D}_{base}$ and PAIP as the novel dataset $\mathcal{D}_{novel}$. Liver tissues from PAIP are different from colon tissues in NCT. We thus regard them as are out-domain novel classes. To study *heterogeneous* and *homogeneous* shot selection problem, we use WSI ID information to split PAIP into a support WSI set (15 WSIs with 22.5k images) and a query WSI set (35 WSIs with 52.5k images). During evaluation, support samples and query samples are drawn from the support WSI set and the query WSI set respectively. *Heterogeneous* strategy selects few-shot samples from different support WSIs, while *homogeneous* strategy selects them from a single randomly chosen support WSI.

**Evaluation.** If not otherwise specified, for near-domain task, we evaluate methods in $1000 \times 9$ (9 sub-tasks) random meta-tasks; for mixture- and out-domain tasks, we evaluate methods in 1000 randomly sampled meta-tasks. All meta-tasks use 15 samples per class as the query set. We report the average F1-score and 95% confidence interval. To handle the unequal numbers of base classes and novel classes, we follow convention in GZSL (Xian et al., 2018) and GFSL (Shi et al., 2020) to report their average harmonic mean. More details about evaluation metrics are in Appendix A.

**Implementations.** **I. Pre-training.** We use ResNet-18 as the embedding function $f_\phi$, and follow previous arts in FSL (Tian et al., 2020; Chen et al., 2019) to use $l_2$-normalized features for clustering and downstream meta-tasks. **II. Latent Augmentation.** We use faiss (Johnson et al., 2019), a library for clustering, to perform K-means with a fixed seed for reproducibility. The number of prototypes in the base dictionary is 16 ($C = 16$, discussed in ablation §4.3). In each meta-task, each sample is augmented 100 times (including the original one) by LA. More details are in Appendix C.

**Compared methods.** Recent works (Chen et al., 2019; Tian et al., 2020), including a concurrent work for histology image (Shakeri et al., 2021), show that sophisticated episodic training (meta-training) is no better than standard pre-training. Hence, we summarize methods using standard pre-training as: 1) NearestCentroid. It computes class centroids from support sets and classifies query samples to their nearest centroids. Related works using such strategy includes Wang et al. (2019a), Snell et al. (2017), and Chen et al. (2020c), to list a few; 2) LinearClassifier. It trains a new fully-connected layer with different loss functions (Chen et al., 2019; Lee et al., 2019) w.r.t. support samples, or directly uses linear models in scikit-learn (Pedregosa et al., 2011), *e.g.*, LogisticRegression (Yang et al., 2021; Tian et al., 2020). For the ease of implementation and consistency, we use NearestCentroid, and two $l_2$-regularized linear classifiers — LogisticRegression and RidgeClassifier, all from scikit-learn APIs (Pedregosa et al., 2011).

## 4.2 MAIN RESULTS

**Fully-supervised $f_\phi^{FSP}$ v.s. Self-supervised $f_\phi^{CLP}$.** Results in Table 1 show that CLP generalizes better to novel classes than FSP by a large margin. Comparing the best vanilla entries (w/o. LA) using two types of pre-training methods, we observe an average advantage in HarmMean of 4%, 5% and 8% in 1-/5-/10-shot settings by CLP in near-domain task and 10%, 19%, 16% in mixture-domain task. Besides, CLP representations benefit more from the increase of number of shots than FSP's in both tasks, e.g. +17% vs. +11% and +12% vs. +10% when 1-shot → 5-shot for linear classifiers in near-domain, and mixture-domain tasks respectively. Despite the inevitable advantage of FSP in base classes under full supervision, CLP demonstrates stronger generalizability to novel classes. Furthermore, Table 3 also confirms the superiority of CLP over FSP in out-domain task where a

Table 1: **Main results in near-/mixture-domain tasks.** In near-domain task, the "Base"/"Novel" columns report average F1-scores of the base/novel classes; the "HarmMean" columns report their average harmonic mean. In mixture-domain task, the same metrics are reported w.r.t. middle-domain classes and out-domain classes. "$\pm$" numbers denote 95% confidence interval across multiple runs. "LA" denotes latent augmentation. More 10-shot results are in Table D.1.

| | | 1-shot | | | 5-shot | | 10-shot |
|---|---|---|---|---|---|---|---|
| 9-way-$K$-shot | | | | Near-domain task | | | |
| Methods | Base | Novel | HarmMean | Base | Novel | HarmMean | HarmMean |
| *Fully-supervised pre-training (FSP)* | | | | | | | |
| NearestCentroid | 77.38±0.96 | **43.80±1.12** | **54.84±1.03** | 88.64±0.41 | 57.67±0.80 | 68.36±0.53 | 71.00±0.46 |
| LogisticRegression | 75.14±1.03 | 37.80±1.17 | 48.84±1.09 | 88.45±0.40 | 48.76±0.93 | 59.99±0.55 | 66.39±0.45 |
| RidgeClassifier | 75.89±1.02 | 37.55±1.18 | 48.75±1.09 | 88.44±0.40 | 45.73±0.97 | 56.96±0.57 | 60.33±0.48 |
| LogisticRegression + LA (ours) | **78.88±0.94** | 43.42±1.14 | 54.83±1.02 | **90.85±0.36** | 63.54±0.74 | 73.63±0.48 | **78.14±0.39** |
| RidgeClassifier + LA (ours) | 76.19±1.03 | 40.71±1.16 | 51.95±1.07 | 88.86±0.41 | 53.90±0.90 | 64.87±0.55 | 66.96±0.46 |
| *Contrastive-learning pre-training (CLP)* | | | | | | | |
| NearestCentroid | 71.45±0.95 | 51.95±1.03 | 58.81±0.98 | 83.11±0.52 | 65.36±0.80 | 72.51±0.62 | 75.18±0.54 |
| LogisticRegression | 70.83±1.01 | 48.76±1.12 | 56.13±1.06 | 84.04±0.50 | 62.69±0.87 | 70.89±0.62 | 76.83±0.51 |
| RidgeClassifier | 71.24±0.99 | 49.18±1.12 | 56.56±1.05 | 85.89±0.46 | 66.12±0.83 | 73.73±0.58 | 79.45±0.45 |
| LogisticRegression + LA (ours) | 72.11±0.95 | 53.15±1.08 | 59.82±1.01 | **86.43±0.46** | 76.68±0.61 | 80.67±0.51 | 85.48±0.40 |
| RidgeClassifier + LA (ours) | **72.60±0.99** | **54.50±1.11** | **60.89±1.04** | 86.18±0.47 | **78.00±0.60** | **81.28±0.51** | **86.17±0.40** |
| 5-way-$K$-shot | | | | Mixture-domain task | | | |
| Methods | Middle | Out | HarmMean | Middle | Out | HarmMean | HarmMean |
| *Fully-supervised pre-training (FSP)* | | | | | | | |
| NearestCentroid | 45.65±1.27 | **54.94±1.22** | 49.87±1.24 | 49.01±1.05 | 61.28±0.78 | 54.56±0.90 | 55.75±0.84 |
| LogisticRegression | 40.07±1.35 | 48.00±1.44 | 43.68±1.39 | 49.42±1.02 | 54.18±1.04 | 51.69±1.03 | 56.12±0.93 |
| RidgeClassifier | 41.46±1.36 | 48.74±1.43 | 44.81±1.39 | 55.28±0.98 | 56.12±1.05 | 55.70±1.01 | 60.77±0.88 |
| LogisticRegression + LA (ours) | 46.98±1.33 | 53.34±1.30 | **49.95±1.31** | 65.51±0.81 | 62.64±0.87 | **64.04±0.84** | **67.60±0.73** |
| RidgeClassifier + LA (ours) | **47.70±1.38** | 52.13±1.35 | 49.82±1.36 | **67.45±0.80** | 60.97±0.95 | 64.04±0.86 | 67.23±0.74 |
| *Contrastive-learning pre-training (CLP)* | | | | | | | |
| NearestCentroid | 71.42±1.14 | 52.01±1.05 | 60.19±1.09 | 84.50±0.49 | 65.31±0.71 | 73.68±0.58 | 76.30±0.49 |
| LogisticRegression | 72.16±1.06 | 51.14±0.97 | 59.86±1.01 | 83.91±0.49 | 61.98±0.71 | 71.29±0.58 | 74.89±0.48 |
| RidgeClassifier | **72.57±1.04** | 51.13±0.96 | 59.99±1.00 | 85.22±0.43 | 62.47±0.72 | 72.09±0.54 | 75.84±0.46 |
| LogisticRegression + LA (ours) | 71.77±1.09 | 52.73±1.03 | 60.79±1.06 | 87.51±0.39 | 72.92±0.65 | 79.55±0.48 | 84.95±0.41 |
| RidgeClassifier + LA (ours) | 71.86±1.08 | **52.92±1.04** | **60.95±1.06** | **88.55±0.38** | **74.04±0.65** | **80.64±0.48** | **86.32±0.39** |

Table 2: **Ablations on covariance type.** See text for more details.

| Cov Type | Base | Novel | HMean |
|---|---|---|---|
| None | 85.85±0.78 | 53.27±1.63 | 65.74±1.06 |
| Tied | 79.35±1.08 | **65.32±1.21** | 71.65±1.14 |
| Diag | 85.91±0.88 | 62.66±1.42 | 72.46±1.08 |
| Spherical | 85.78±0.87 | 62.00±1.39 | 71.97±1.07 |
| Full (default) | **87.51±0.80** | 65.79±1.36 | **75.11±1.01** |

Table 3: **Results in out-domain tasks.** Average F1-scores from 1000 meta-tasks are reported.

| RidgeClassifier | Homogeneous | | | Heterogeneous | | |
|---|---|---|---|---|---|---|
| 3-way $K$-shot | FSP | CLP | CLP+LA | FSP | CLP | CLP+LA |
| $K = 1$ | 36.90 | 42.56 | **43.14** | / | / | / |
| $K = 5$ | 39.00 | 48.91 | 49.83 | 43.35 | 52.25 | **53.67** |
| $K = 10$ | 40.26 | 50.57 | 51.62 | 45.91 | 55.96 | **58.35** |
| $K = 50$ | 41.53 | 51.76 | 53.71 | 50.54 | 61.88 | **65.38** |
| $K = 100$ | 41.23 | 52.74 | 54.25 | 52.45 | 64.03 | **67.56** |

larger domain shift exists. Such generalization gap between FSP and CLP in histology images is at slight odds with observations in natural images, where they show similar generalizability. We study and discuss it at §4.4. We also provide the linear evaluation results of all the 20 pre-trained models in Appendix C.3 for a reference to see how each model perform in NCT dataset.

**Latent augmentation brings consistent improvement.** Regardless of pre-training methods, LA brings consistent gains over baseline linear classifiers, confirming its effectiveness. With base dictionaries, limited few-shot samples are able to proliferate in a reasonable way by transferring latent variations. Such boost maintains its significance from near-domain task to mixture-domain task (Table 1) but turns smaller in out-domain task (Table 3). This is in our expectation since the three classes (non-tumor, viable-tumor and other) defined in PAIP are extremely coarse-grained: it may include couples of real fine-grained classes (*c.f.* Figure A.4 in Appendix). Few samples are unable to well represent their entangled semantics. Hence, this observation does not repudiate the effectiveness of latent augmentation but re-ensures its tenability.

### 4.3    ABLATIONS

To ablate design choices, we exclude two cancer-related classes, *i.e.*, cancer-associated stroma (STR) and colorectal adenocarcinoma epithelium (TUM), from NCT to be novel classes, and use the rest as base classes. If not otherwise specified, all ablations are conducted on CLP models with Ridge-Classifier for 300 meta-tasks in 5-shot setting.

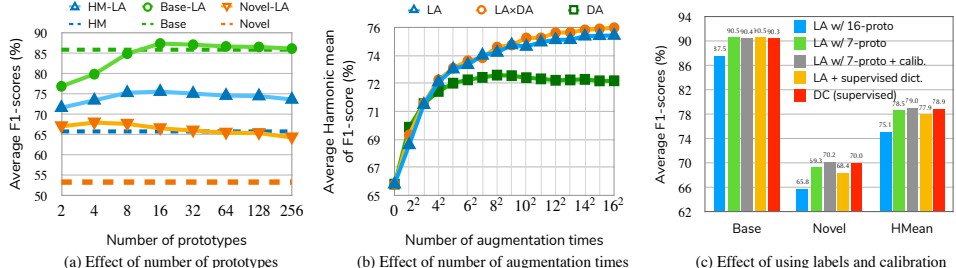

Figure 3: **Ablations on latent augmentation.** (a) The effect of different number of prototypes. Dash lines are the baselines for the solid lines of same colors. (b) The effect of number of augmentation times. The harmonic mean are plotted. "LA×DA" denotes $T$ latent augmentations are applied after $T$ traditional data augmentations (leads to $T^2$ times in total). (c) The effect of using labels and calibration. "DC": Distribution Calibration. "calib.": calibration; we introduce it in Appendix D.3.

**Number of prototypes in base dictionary.** Figure 3 (a) shows how performance varies with the number of prototypes $C$. We observe the similar tendency between base class and novel class, where their harmonic means peak at $C = 16$; we subsequently choose $C = 16$ for all experiments. Besides, the performance of base classes and novel classes shows opposite trends from $C = 4$ to $C = 16$. The trade-off exists here that as the granularity of clusters increases ($C \uparrow$), the intra-cluster variance decreases, which results in better grouping accuracy but brings less semantic variation. The novel classes benefit from larger variation while the base classes benefit from more accurately estimated variation since they have been exposed in training. Nevertheless, LA demonstrates its robustness by consistent improvement over baselines (solid vs. dashed lines of same color in Fig. 3-(a)).

**DA vs. LA, and number of augmentation times.** Here we compare LA with data augmentation (DA), and their combination. DA's details are in Appendix D.2. Figure 3-(b) shows that LA outperforms DA by a large margin. The boost brought by DA saturates easily and keeps dropping thereafter, while LA keeps improving with all tested cases. Besides, DA can marginally improve LA (LA×DA v.s. LA). We conclude that LA has already covered the role played by DA in an implicit way since the most of gains are brought by LA. It is worthy to emphasize the computation budget involved in LA (addition in $\mathbb{R}^d$ space) is significantly lower than DA (image augmentation in $\mathbb{R}^{3HW}$ space and encoder forwards). Therefore, we run all experiments only with LA.

**Using label information.** LA constructs the base dictionary without *any* label information, *e.g.*, the number of classes. When label is available, similar methods such as Distribution Calibration (DC) (Yang et al., 2021) can be used. Figure 3-(c) shows the comparisons of using labels and calibration (introduced in Appendix D.3). Under supervision, "DC" and "LA+supervised dict." achieve competitive performance. Surprisingly, once given the number of base classes, "LA w/ 7-proto" can attain better results than using 16-prototype and be comparable as supervised DC. Calibration could further improve LA. This implies that, with LA, knowing the number of base classes can be sufficient for gaining as descent results as knowing all examples' labels.

**Covariance type.** Here we explore more types of covariances that LA can use. Specifically, we also include: 1) "Tied", where all clusters share a covariance matrix estimated from the entire base dataset, 2) "Diag", where each cluster has its own diagonal covariance matrix, *i.e.*, diagonal elements are a variance vector and non-diagonal elements are zeros, 3) "Spherical", where each cluster has its own single scalar-variance shared by all feature dimensions. Table 2 shows the results. LA with all types of covariances improves performance. This emphasizes the importance of diversifying few samples with variation. Using full covariance estimation achieves the best performance. We further show how different covariance types perform with smaller or larger cluster sizes in Appendix D.4.

**Heterogeneous v.s. Homogeneous patch selection.** We investigate the hetero-/homo-geneous patch selection strategies defined in out-domain task (§4.1). Table 3 shows the results. We observe: i) heterogeneous selection shows higher baselines than homogeneous one; and ii) LA brings more gains for heterogeneous selection. Heterogeneous patches provide reliable and diverse "anchor" samples than homogeneous ones, thus can benefit more from bootstrapping the base dictionary.

**Additional ablation studies.** We conduct two more ablation studies: (a) clustering with different random seeds in K-Means, (b) studying the effect of $l_2$-normalization to covariance. We show that LA's improvement is stable under different random seeds used in K-Means (Appendix D.5),

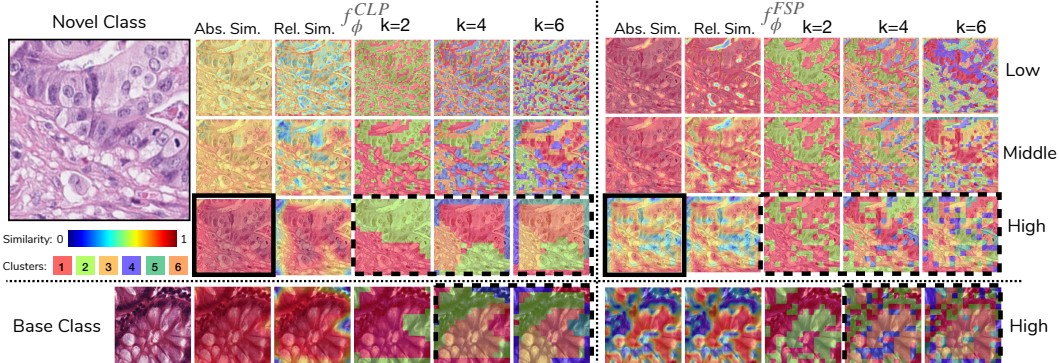

Figure 4: **Visualization of samples learned by CLP and FSP.** "Abs./Rel. Sim." columns show the absolute/relative cosine similarity between the global feature and the local ones. Relative similarity is the min-max normalized absolute similarity. $k$ indicate the cluster numbers used by K-Means. "Low", "middle" and "high" denote using features from stage-3, 4, and 5 from a ResNet. See text in §4.4 for discussion. Visualization procedures and more examples are provided in Appendix D.8.

and covariances estimated before and after $l_2$-normalization are highly correlated, and further $l_2$-normalizing augmented samples marginally degenerates LA's performance (Appendix D.6). In addition, we reproduce $\delta$-encoder (Schwartz et al., 2018) for ablation task. However, we observe performance drops from baseline (*e.g.*, -2.5% HMean), and none of the tested cases can outperform our method. More results and discussions are in Appendix D.7.

## 4.4 MORE DISCUSSION

**Disparity between $f_\phi^{CLP}$ and $f_\phi^{FSP}$ influences the choice of base learner.** In Table 1, we find i) stronger baselines for CLP and FSP vary, and ii) simple NearestCentroid can sometimes outperform the vanilla $l_2$-regularized linear classifiers for FSP. Here we briefly discuss our understandings. Representations produced by CLP can have different distributions compared to FSP, as also noticed by He et al. (2020). With limited training samples, different classifiers can have their own biases in building decision boundary, leading to different generalizability. Besides, no regularization techniques are used during FSP (Chen et al., 2020a), *e.g.*, weight decay (Krogh & Hertz, 1992), DropBlock (Lee et al., 2019; Tian et al., 2020) or "distill" regularization (Tian et al., 2020). The linear classifiers, though with $l_2$ penalty, may still be overfitted in such representation space when only limited samples are provided. Subsequently, the simplest NearestCentroid model can yield better results than these overfitted linear models, as it has the least complexity.

**Why do CLP models generalize better than FSP ones in histology images?** To study why the large generalization gap exists, in Figure 4, we follow Chen & Li (2020) to see how features aggregate in space. Specifically, we visualize the cosine similarity between a feature map (a set of local representations) and its global averaging (global representation), and run K-Means on the feature maps from different layers (*i.e.*, stage 3, 4, 5 of ResNet) with different cluster numbers. We observe: the FSP model maintains high global-local similarity in low-/middle-levels, while the CLP model holds it in high-level (solid boxes). Besides, the CLP model extracts low-level features that are edge-related, and afterwards successively agglomerates adjacent similar structures (dashed boxes). In contrast, the FSP model can differentiate nuclei in low-&middle-levels but fails to encode structure-related features in a deeper layer.

We further visualize some samples from base classes (bottom of Figure 4 and D.6), and find that such disparity between FSP and CLP does not only exist in a previously unseen class but also in seen classes. In the bottom row of Figure 4, FSP only pays attention to the most discriminative parts, leaving the rest "redundant parts" disorder (dashed box). However, the discriminative parts are likely to alter when a new class is presented. FSP's inability to fully encode meaningful information may lead to its failure in generalizing to new classes. Meanwhile, CLP encodes most of tissue-structure-related features that may be useful for novel class recognition, possibly resulting in better generalizability. However, FSP models and CLP models are shown to perform similarly, instead of differently, under the same visualization procedure in ImageNet dataset (see the website for a comparison). What might cause this disparity? ImageNet has more diverse classes (1000 classes) and samples ($\sim$ 1.28M images) than those in histology datasets. FSP models in ImageNet need to recognize the discriminative parts of all 1000 classes. In such case, the redundant information

in one class might contribute to the recognition of another class. Therefore FSP may eventually encode most of available information for new classes that are related to ImageNet classes. However, histology datasets usually lack enough diverse annotated classes that help to build a know-everything FSP model. A topic we leave for future work is to explore whether CLP always generalizes better than FSP when pre-training on a base dataset with limited number of annotated classes and if the generalization gap would increase as the label diversity decreases. We point out that the visualization results and the large generalization gap shown in our work still remains as empirical observations. Our discussion is about what the reasons behind them could be. We hope our work would be helpful for representation learning, histology image analysis, and beyond.

## 5 RELATED WORK

**Few-shot learning (FSL).** FSL has been tackled from different perspectives, *e.g.* metric-based and optimization-based (Finn et al., 2017; Rusu et al., 2018). This paper follows a "pre-training & fine-tuning" scheme in metric-based branch, where previous works typically learn a shared metric space by standard fully-supervised pre-training (Tian et al., 2020; Chen et al., 2019; Wang et al., 2018). In contrast, we propose to incorporate self-supervised pre-training to enable label-efficient learning, and show that it can yield stronger generalization than supervised pre-training.

**FSL in medical images.** FSL in medical images is at its early stage, especially for histology images. Mahajan et al. (2020) investigate FSL methods in skin-disease classification, while Chen et al. (2021a) tackle COVID-19 CT image classification using contrastive pre-training and prototypical network fine-tuning. In the regard of histology image, Medela et al. (2019) use a triplet loss (Schroff et al., 2015) to pre-train an encoder with a followed fine-tuned SVM classifier for few-shot domain adaptation. Sikaroudi et al. (2020) and Teh & Taylor (2020) study learning with less data in histology images. Concurrent to our work, Shakeri et al. (2021) simultaneously propose a benchmark for few-shot classification of histological images. Our work has similar but different settings, with more investigations conducted, *e.g.*, GFSL task, and hetero-/homo-geneous few shots selection.

**Self-supervised learning.** Self-supervised learning aims to learn good representations without true labels. Recent state-of-the-art variants can be categorized as contrastive-based learning (Chen et al., 2021b; 2020a; He et al., 2020), cluster-based learning (Caron et al., 2018; 2020) and expectation prediction based learning (Grill et al., 2020; Chen & He, 2021). However, this line of works focus on pre-training on ImageNet-like images, and recent attention has been attracted to images with *multi-objects* and *multi-texture* presented (Chen & Li, 2020). We see histology image as a natural choice for such study, and show that contrastive learning can agglomerate structural "part-whole" information and maintain "global-local consistency", which make it generalize better for such data than supervised counterparts (see §4.4 and Table C.1 in Appendix C.3).

**Representation variation augmentation.** The idea of exploiting feature variations dates back to a decade ago (Heller et al., 2009; Salakhutdinov et al., 2012). Recent variants further develop this idea. For example, Hariharan & Girshick (2017) and Schwartz et al. (2018) use a generator to generate "hallucinated" novel features from variation of given base samples. This method is later extended by not relying on given base samples (Wang et al., 2018). Wang et al. (2019b) use class variance to perform semantic augmentation for classification and segmentation (Wang et al., 2021), while Yin et al. (2019) and Liu et al. (2020) utilize intra-class variance of head classes to augment tail classes for "long-tail" face recognition problem. Yang et al. (2021) use the distribution information, i.e. mean and variance, of base classes to calibrate novels' distribution. Cheung & Yeung (2020) propose a novel and systematical framework to apply automated augmentation with more considerations in label-preserving transformations. This work follows the line of Yang et al. (2021); Wang et al. (2021); Liu et al. (2020), but in contrast to them, we obtain and transfer variations without relying on any label information, allowing our method to scale gracefully to other label-hungry problems.

## 6 CONCLUSION

This work has studied, as an early attempt, the few-shot learning problem for histology images. We incorporate contrastive learning and latent augmentation to fully exploit training data in an unsupervised way, which means our method can gracefully scale to other large label-hungry problems. More importantly, we show that the generalization gap between the state-of-the-art contrastive learning pre-training method and supervised pre-training in histological images is larger than that in ImageNet experiments. We analyze the underlying reasons and provide our empirical understandings. We hope our work could contribute to the study of representation learning and generalization for both self-supervised learning community as well as histology image analysis community.

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

# A  DATASET

## A.1  NCT

**Data.**  NCT-CRC-HE-100K dataset (Kather et al., 2018) contains $100,000$ non-overlapping patches extracted from hematoxylin and eosin (H&E) stained human colorectal cancer and normal tissues. Each image is of size $224 \times 224$ at 0.5 MPP ($20\times$ magnification). Tissue classes and their label index are: 0) background (BACK), 1) adipose (ADI), 2) debris (DEB), 3) lymphocytes (LYM), 4) mucus (MUC), 5) smooth muscle (MUS), 6) normal colon mucosa (NORM), 7) cancer-associated stroma (STR), 8) colorectal adenocarcinoma epithelium (TUM). Figure A.2 shows the class distribution and Figure A.1 shows 4 examples per class from NCT. These images are from National Center for Tumor Diseases (Heidelberg, Germany) and University Medical Center Mannheim (Mannheim, Germany). Their acquisition protocols differ across organizations, which lead to inter-source domain shift.

**Processing.**  We use the "NCT-CRC-HE-100K-NONORM" dataset, which does not apply color normalization to images. We randomly split NCT dataset (100k images) into a training set (80k images) and a test set (20k images). The class distribution is the same for the training set and test set, i.e. data are sampled w.r.t. each class. Since each image in NCT is of size $224 \times 224$, we do not further resize it.

**Near-domain task.**  NCT dataset contains tissue patches from multiple sources, which bears slight domain shift since the staining intensity varies. To better study the generalizability in FSL problem, we do leave-one-class-out-as-novel-class to the training set of NCT. This procedure closely resembles the leave-one-out cross-validation. Since histology datasets usually have far less number of classes compared to natural image dataset (e.g. 9 classes in NCT v.s. 100 classes in *mini*ImageNet and 608 classes in *tiered*ImageNet), we believe such leave-one-class-out-as-novel-class can better simulate the generalization error bound when number of classes is small. After the leave-one-class-out-as-novel-class split, 9 sub-base datasets are constructed, each of which has one class regarded as the novel class and samples belonging to it are excluded from pre-training. For each sub-base dataset, a CLP model and a FSP model will be trained on it. Therefore, a total of 18 models will be obtained (9 CLP models and 9 FSP models). All models are evaluated on the NCT test set. It contains both seen classes and unseen novel classes, so as to simulate generalizes few-shot classification.

**Evaluation metric.**  Typical FSL methods use accuracy as metrics. However, to study GFSL problem, the performance of novel classes and base classes should be appropriately considered. Accuracy w.r.t. specific classes cannot be computed in a joint label space. Hence, we choose F1-score as our metric. Besides, the numbers of novel classes and base classes are unequal in each sub-task, so we follow convention in GFSL to compute their harmonic mean. The final metrics are

$$M_{base}(i) = \frac{1}{8} \sum_{j \in \{0..9\} \setminus \{i\}} F_1(j), \tag{A.1}$$

$$M_{novel}(i) = F_1(i), \tag{A.2}$$

$$HMean(i) = \frac{2}{1/M_{base}(i) + 1/M_{novel}(i)}, \tag{A.3}$$

$$\text{Base} = \frac{1}{9} \sum_{i=1}^{9} M_{base}(i), \quad \text{Novel} = \frac{1}{9} \sum_{i=1}^{9} M_{novel}(i), \quad \text{HarmMean} = \frac{1}{9} \sum_{i=1}^{9} HMean(i), \tag{A.4}$$

where $F_1(j)$ denotes the F1-score of $j$-th class. $M_{base}(i)$ (Equation A.1) computes the average F1-score of base classes in $i$-th sub-task. Equation A.2 and Equation A.3 compute the F1-score of novel class and the harmonic mean of $M_{base}$ and $M_{novel}$ in $i$-th sub-task respectively. Equation A.4 shows the final metrics we report in this work.

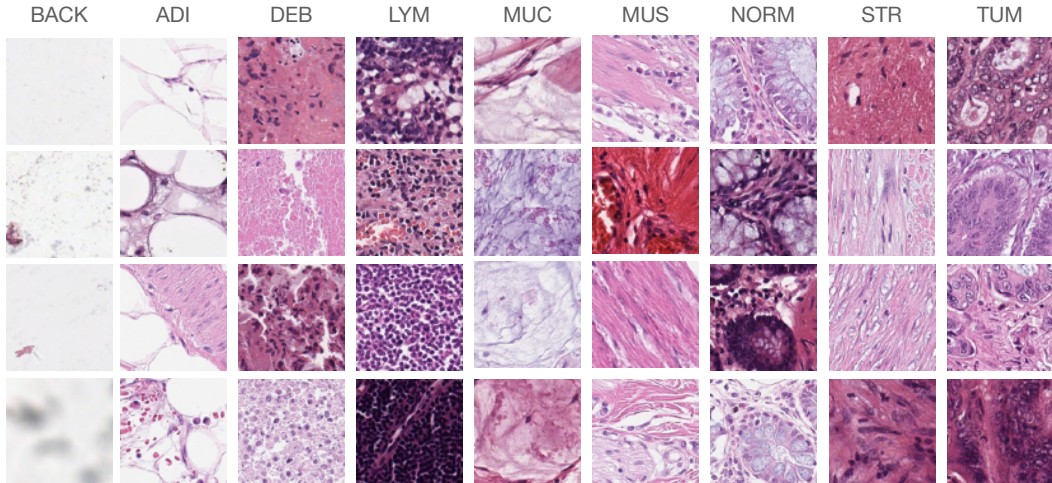

Figure A.1: Example images from NCT dataset. Each column represents a different class.

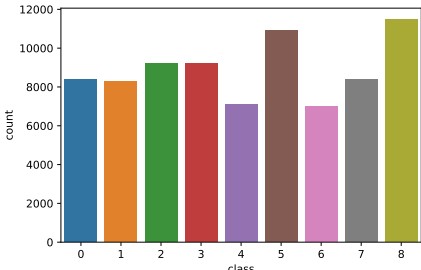

Figure A.2: Class distribution in NCT dataset. Labels denote: 0-ADI, 1-BACK, 2-DEB, 3-LYM, 4-MUC, 5-MUS, 6-NORM, 7-STR and 8-TUM. See Appendix A.1 for full names of labels.

## A.2 LC25000

**Data.** The LC25000 (LC-25K) dataset contains 25,000 color images from 5 classes. They are: 0) colon adenocarcinoma (Colon_ACA), 1) benign colonic tissue (Colon_benign), 2) lung adenocarcinoma (Lung_ACA), 3) benign lung tissue (Lung_benign), and 4) lung squamous cell carcinoma (Lung_SCC). The class distribution is balanced, i.e. each class has 5000 images. All images are of size $768 \times 768$. LC25K is constructed by augmenting 1250 images (250 images for each class). The augmentations are: left and right rotations (within 25 degrees, p=1.0) and vertical and horizontal flips (p=0.5), where "p" represents the probability.

**Processing.** We resize all images from $768 \times 768$ to $224 \times 224$. No other process is taken.

**Mixture-domain task.** LC-25K dataset includes tissues from colon and lung sites. Two of five classes in LC-25K are colonic tissues, which suffers moderate domain shift due to difference in imaging protocol and pixel resolutions. Thus, we see them as middle-domain novel classes. The rest three lung-related classes are regarded as out-domain novel classes since they are from a different organ. To study few-shot classification, we use the entire training set (80k images) of NCT as $\mathcal{D}_{base}$, and use LC-25K as $\mathcal{D}_{novel}$ for evaluation.

**Evaluation metric.** Similar to near-domain task evaluation in Appendix A.1, we compute $\text{Middle} = \frac{1}{2}(F_1(0) + F_1(1)), \text{Out} = \frac{1}{3}(F_1(2) + F_1(3) + F_1(4)), \text{HarmMean} = \frac{2}{1/\text{Middle}+1/\text{Out}}$. Here $F_1(i)$ denotes the F1-score of $i$-th class. Class 0 and class 1 are colon-related, termed as middle-domain novel classes, and the rest three classes are lung-related, termed as out-domain novel classes.

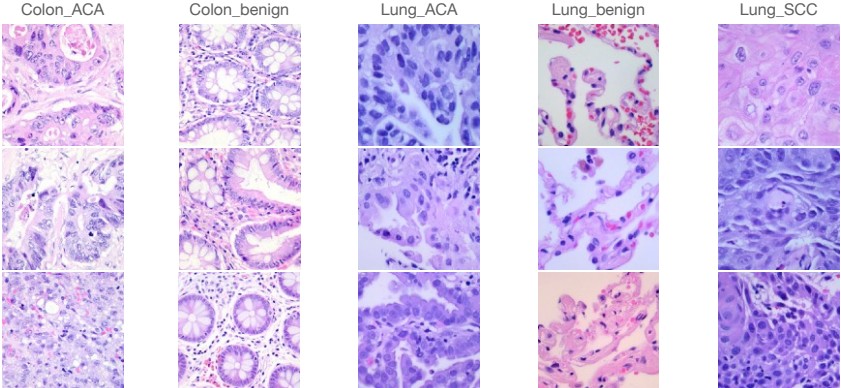

Figure A.3: Example images from LC25000 dataset. Each column represents a different class. The classes are: colon adenocarcinoma (Colon_ACA), benign colonic tissue (Colon_benign), lung adenocarcinoma (Lung_ACA), benign lung tissue (Lung_benign), and lung squamous cell carcinoma (Lung_SCC).

### A.3 PAIP19

**Data.** The PAIP 2019 (Kim et al., 2021) is composed of 50 H&E stained WSIs at $20\times$ magnification. Each WSI is approximately of size $45,000 \times 45,000$ with XML annotation. It contains three classes, which are: 0) non-tumor liver tissue, 1) viable tumor, and 2) other tissues in whole tumor area but not viable tumor. The class 2 may include intratumoral hemorrhage, necrosis or non-tumor tissue in whole tumor area. Figure A.4 shows an example WSI of PAIP19. This dataset provides "case number" (WSI identifier) information.

**Processing.** We use mask information to extract 500 randomly selected non-overlapping patches at $20\times$ magnification for each class from each WSI, leading to a dataset with $75,000$ samples. We define patches from 15 randomly selected WSIs as the support WSI set and patches from the rest 35 WSIs as query WSI set.

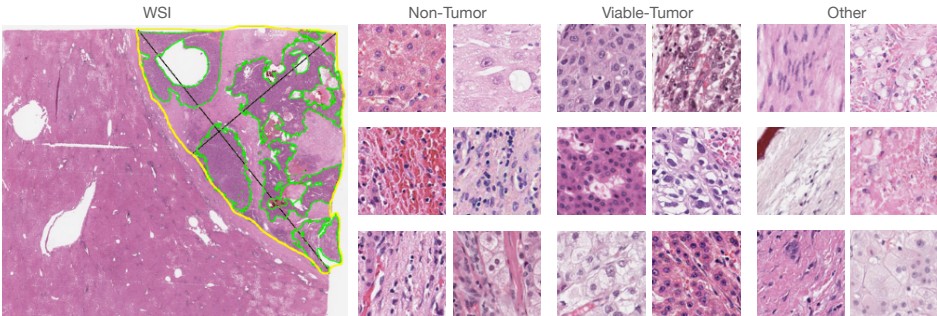

Figure A.4: Examples from PAIP 2019 dataset. The Leftmost image shows the overview of a whole-slide image, where tissues outside yellow boundary are defined as "non-tumor" class, and tissues inside green boundary are "viable-tumor" class, and tissues inside yellow but outside green are "other" class. Right columns show examples of extracted patches from PAIP19 dataset. The "non-tumor" class and "other" class are coarse-defined, since they can include many real fine-grained classes such as hepatocyte, vein, artery, stroma, fibrosis, fatty infiltration, etc.

## B CONTRASTIVE LEARNING PRE-TRAINING: MOCO-V3

MoCo-v3 (Chen et al., 2021b) aims for an empirical study for vision transformer pre-training, but has a ResNet variant. We adopt the ResNet variant.

**Overview.** MoCo-v3 follows its ancestors MoCo v1/2 (He et al., 2020; Chen et al., 2020b) with straightforward modification. As shown in Figure B.1, each batch is augmented twice under stochastic augmentation $t$ to obtain two views, denoted as $x_1, x_2 \sim t(x)$, where $x$ represents the raw images. They are then encoded by query encoder and key encoder respectively. The query encoder is composed of three components: a backbone $f_\phi$, a projector (projection head) $f_g$ and a predictor (prediction head) $f_q$, formulated as $f_q \circ f_g \circ f_\phi$. The key encoder, also known as momentum encoder, consists of momentum copies of backbone and projector, formulated as $f_{\tilde{g}} \circ f_{\tilde{\phi}}$.

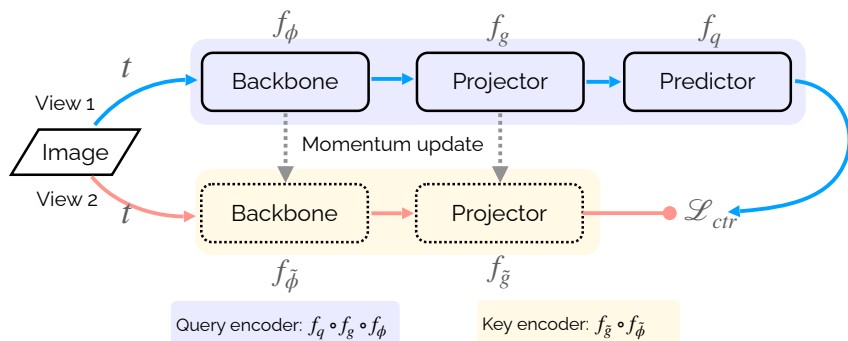

Figure B.1: Abstraction of MoCo-v3 structure. Each image is augmented twice under stochastic augmentation $t$ to obtain two views (denoted by blue and red color). The key encoder (bottom, $f_{\tilde{g}} \circ f_{\tilde{\phi}}$ is momentum updated by parts query encoder (top, $f_q \circ f_g \circ f_\phi$). The ˜ symbol denotes momentum updated parameters.

**Momentum update.** In iteration $k$, the momentum update rule is:

$$\tilde{\phi}_k \leftarrow m\tilde{\phi}_k + (1-m)\phi_k, \qquad \tilde{g}_k \leftarrow m\tilde{g}_k + (1-m)g_k, \tag{A.5}$$

where $m$ is the momentum.

**Loss function.** The contrastive loss function is a form of InfoNCE (Oord et al., 2018):

$$\mathcal{L}_{ctr}(u, v) = -\log \frac{\exp(u \cdot v^+/\tau)}{\exp(u \cdot v^+/\tau) + \sum_{v^-}\exp(u \cdot v^-/\tau)}, \tag{A.6}$$

Here, $v^+$ denotes the positive sample, i.e. the other augmented view of $u$, and $v^-$ denotes the negative samples, i.e. other representations in the batch. $\tau$ is a hyper-parameter, known as temperature (Wu et al., 2018). The final loss is a symmetric sum: $\mathcal{L}_{CLP} = [\mathcal{L}_{ctr}(z_1, \tilde{z}_2) + \mathcal{L}_{ctr}(z_2, \tilde{z}_1)]/2$, where the subscript indicates view source, $z = l_2[f_q \circ f_g \circ f_\phi(x)]$, $\tilde{z} = l_2[f_{\tilde{g}} \circ f_{\tilde{\phi}}(x)]$, and $l_2[\cdot]$ means $l_2$ normalization.

## C    IMPLEMENTATION DETAILS

### C.1    FULLY-SUPERVISED PRE-TRAINING

**Optimization.** We use SGD optimizer with $lr = 0.5, momentum = 0.9$, and no weight decay is used, i.e. weight_decay=0. The batch size is 512. We train for 100 epochs with "step decay" learning schedule. The $lr$ is multiplied by 0.1 at 30, 60 and 90 epochs respectively.

**Data augmentation.** We use `RandomResizedCrop`, `RandomHorizontalFlip`, followed by normalization (subtract `mean` and divide `std`) using ImageNet (Deng et al., 2009) statistics, i.e. $mean = (0.485, 0.456, 0.406)$ and $std = (0.229, 0.224, 0.225)$. During testing, we only resize the image to $224 \times 224$, followed by ImageNet normalization. Following previous work (Chen et al., 2019; Tian et al., 2020), we further $l_2$-normalize features.

### C.2    CONTRASTIVE-LEARNING PRE-TRAINING

The detailed description of MoCo-v3 is in Appendix B.

Table C.1: The linear evaluation accuracy of FSP and CLP models. FSP means fully-supervised pre-training, while CLP denotes contrastive-learning pre-training. We train LogisticRegression models on whole training features, and report the accuracy for whole test features. Base-Base means the novel class is excluded when we train and test linear classifiers. Joint-Joint means the linear classifiers are trained on all training features (80k samples) and then evaluated on all test features (20k samples). The "Average" row reports the average accuracy of 9 sub-tasks (the top 9 rows).

| | Base-Base | | Joint-Joint | |
|---|---|---|---|---|
| Novel Class | FSP | CLP | FSP | CLP |
| 0 | **98.46** | 95.81 | **97.73** | 95.98 |
| 1 | **97.80** | 95.79 | **97.41** | 95.72 |
| 2 | **98.59** | 96.28 | 90.60 | **95.10** |
| 3 | **96.67** | 95.85 | 93.92 | **95.88** |
| 4 | 95.50 | **96.49** | 92.66 | **95.67** |
| 5 | **97.60** | 97.39 | 92.43 | **94.33** |
| 6 | **98.72** | 96.09 | 94.75 | **95.22** |
| 7 | **98.49** | 97.68 | 92.66 | **95.52** |
| 8 | **97.44** | 95.75 | 92.63 | **94.80** |
| Average | **97.80** | 96.35 | 93.87 | **95.36** |
| No Novel Class | **98.43** | 96.00 | / | / |

**Architecture.** The architecture of MoCo-v3 is `ResNet18-Projector-Predictor`. We use the codebase of OpenSelfSup[1]. For projector, we use 3-layer `NonLinearNeckSimCLR` (Chen et al., 2020a), with the dimension transitions of $512 \rightarrow 1024 \rightarrow 1024 \rightarrow 256$. For predictor, we use 2-layer `NonLinearNeckSimCLR`, with the dimension transitions of $256 \rightarrow 1024 \rightarrow 256$. The base momentum $m$ to update key encoder is 0.996 and is linearly increased to 1 as training iteration goes. The temperature $\tau$ for contrastive loss is set to 1.

**Optimization.** We follow Chen et al. (2021b) to use `LARS` optimizer with initial learning rate of 0.3, weight decay of $1.5e - 6$, $momentum = 0.9$, and use `CosineAnnearling` learning schedule. We train all models with batch size of 256 for 200 epochs.

**Data augmentation.** Following previous work (Grill et al., 2020; Chen et al., 2021b), we use strong data augmentation: `RandomResizedCrop`, `RandomHorizontalFlip`, and `ColorJitter` with (brightness=0.4, contrast=0.4, saturation=0.4, hue=0.4) and probability of 0.8, `RandomGrayscale` with probability of 0.2, `GaussianBlur` with probability of 0.5, and `Solarization` with probability of 0.2.

## C.3 LINEAR EVALUATION FOR PRE-TRAINING

In §4.1, we manually construct 9 sub-tasks in near-domain task, thus resulting 9 pre-trained models for each pre-training method (FSP and CLP). Besides, for mixture-domain and out-domain tasks, we use models pre-trained on the entire NCT training dataset with no class excluded.

Here we report their performance w.r.t. linear classifier. Specifically, we use the pre-trained models to extract features from NCT training set and test set. Then, we train LogisticRegression models on whole training features, and report the accuracy for whole test features. We report the results of Base-Base and Joint-Joint. Base-Base means the novel class is excluded when we train and test the linear classifiers, while Joint-Joint means the linear classifiers are trained on all training features (80k samples) and then evaluated on all test features (20k samples).

Table C.1 shows the results. With full supervision, FSP models can achieve better results in base label space (Base-Base). However, when evaluated in joint label space (Joint-Joint), they are worse than CLP models. Besides, in few-shot setting, CLP models underperform FSP models in base label space (*c.f.* the "Base" column in Table 1 near-domain task), but when the number of training samples

---

[1]https://github.com/open-mmlab/OpenSelfSup

Table D.1: Results of 10-shot setting in near-domain task and mixture-domain task. In near-domain task, the "Base" and "Novel" columns report the average F1-score of the base/novel class respectively; the "HarmMean" columns report their average harmonic mean. In mixture-domain task, the same metrics are reported w.r.t. middle-domain and out-domain novel classes. "LA" denotes latent augmentation. The bold numbers denote the best while the underscored numbers denote the second best. Results in this table are in complementary to Table 1.

| | 10-shot Near-domain Task | | | 10-shot Mixture-domain Task | | |
|---|---|---|---|---|---|---|
| Methods | Base | Novel | HarmMean | Middle | Out | HarmMean |
| *Fully-supervised pre-training (FSP)* | | | | | | |
| NearestCentroid | 90.08±0.35 | 60.96±0.72 | 71.00±0.46 | 50.63±0.96 | 62.04±0.75 | 55.75±0.84 |
| LogisticRegression | 90.65±0.32 | 55.92±0.78 | 66.39±0.45 | 55.42±0.87 | 56.84±1.00 | 56.12±0.93 |
| RidgeClassifier | 90.14±0.34 | 49.62±0.86 | 60.33±0.48 | 62.72±0.78 | 58.93±1.01 | 60.77±0.88 |
| LogisticRegression + LA (ours) | **92.75±0.29** | **69.10±0.63** | **78.14±0.39** | 70.10±0.69 | 65.26±0.78 | **67.60±0.73** |
| RidgeClassifier + LA (ours) | 90.69±0.33 | 56.65±0.80 | 66.96±0.46 | **71.09±0.66** | 63.77±0.84 | 67.23±0.74 |
| *Contrastive-learning pre-training (CLP), ours* | | | | | | |
| NearestCentroid | 85.15±0.45 | 68.18±0.72 | 75.18±0.54 | 85.97±0.40 | 68.58±0.65 | 76.30±0.49 |
| LogisticRegression | 87.20±0.41 | 69.83±0.71 | 76.83±0.51 | 85.82±0.38 | 66.42±0.66 | 74.89±0.48 |
| RidgeClassifier | 89.02±0.36 | 73.03±0.66 | 79.45±0.45 | 87.04±0.36 | 67.19±0.66 | 75.84±0.46 |
| LogisticRegression + LA (ours) | **89.62±0.36** | 82.40±0.48 | 85.48±0.40 | 90.44±0.34 | 80.08±0.51 | 84.95±0.41 |
| RidgeClassifier + LA (ours) | 89.45±0.36 | **83.81±0.47** | **86.17±0.40** | **91.77±0.32** | **81.47±0.50** | **86.32±0.39** |

Table D.2: Ablation on the effect of the number of prototypes in the base dictionary. This table shows the numerical results of Figure 3-(a).

| | 5-shot ablation task | | |
|---|---|---|---|
| Number of prototypes | Base | Novel | HarmMean |
| No Latent Augmentation | 85.85 | 53.27 | 65.74 |
| 2 | 76.84 | 66.99 | 71.58 |
| 4 | 79.83 | **67.92** | 73.39 |
| 8 | 84.87 | 67.59 | 75.25 |
| 16 | **87.37** | 66.51 | **75.53** |
| 32 | 87.08 | 65.94 | 75.05 |
| 64 | 86.65 | 65.41 | 74.55 |
| 128 | 86.52 | 65.32 | 74.44 |
| 256 | 86.13 | 64.25 | 73.60 |

increases (no longer few-shot setting), CLP models can achieve similar results as FSP models (Table C.1 "Base-Base" column).

# D    DETAILED RESULTS AND MORE ABLATIONS

## D.1    10-SHOT RESULTS

Table D.1 shows the results of 10-shot settings in near-domain and mixture-domain task, in complementary to results in Table 1.

## D.2    ABLATION ON DATA AUGMENTATION

For ablation study, we exclude cancer-related classes, i.e. cancer-associated stroma (STR) and colorectal adenocarcinoma epithelium (TUM), from NCT training set, and use them as novel classes. An additional CLP model is trained on this base dataset for ablation. Table D.2 shows the numerical results of Figure 3-(a), which studies the effect of the number of prototypes in the base dictionary.

Figure D.1 shows the full results of comparison between data augmentation (DA), latent augmentation (LA), and their combination (LA× DA). It can be seen that, DA brings slight gain when the number of augmentation times is small, and brings negative gain when the number of augmentation times increases. In contrast, LA and LA×DA can consistently improve the baselines for base class and novel class, resulting in best performance in harmonic mean.

The data augmentations used in this experiment are `RandomResizedCrop(scale=(0.8, 1.0))` with probability 1.0, `RandomHorizontalFlip` with probability 0.5, and `ColorJitter(brightness=0.4, contrast=0.4, saturation=0.4, hue=0.2)` with probability 0.8. These augmentations are applied sequentially and jointly on original images.

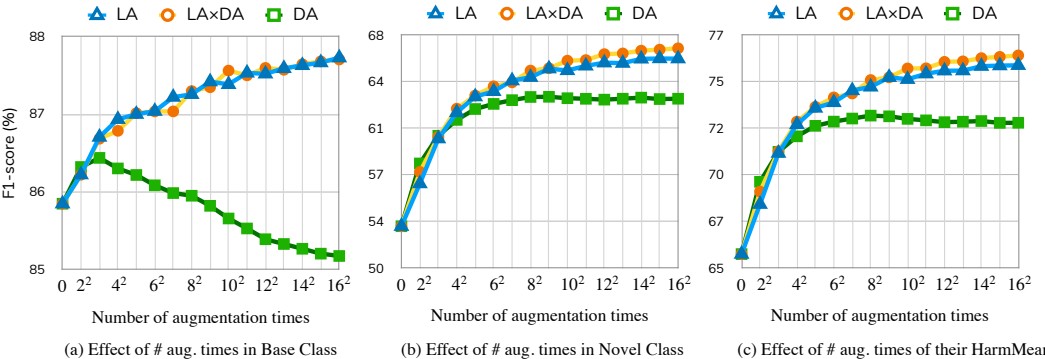

(a) Effect of # aug. times in Base Class     (b) Effect of # aug. times in Novel Class     (c) Effect of # aug. times of their HarmMean

Figure D.1: Comparison of DA, LA and LA×DA. "LA×DA" denotes $T$ latent augmentation applied after $T$ traditional data augmentation (leads to $T^2$ times in total).

### D.3 ABLATION ON USING LABEL INFORMATION AND CALIBRATION

**Supervised base dictionary.** LA constructs the base dictionary using the unsupervised K-Means clustering. Alternatively, it can use label information, if available, to construct the base dictionary. Specifically, the prototypes and their covariance matrices are computed with respect to each class instead of each cluster.

**Calibration.** Distribution Calibration (Yang et al., 2021), a recent state-of-the-art in few-shot classification, proposes to use the statistics of base classes to calibrate the statistics of novel classes. Specifically, they compute a calibrated novel class distribution: $\mu' = (\sum_{i \in \mathbb{S}} \mu_i + z)/(k+1)$, $\Sigma' = (\sum_{i \in \mathbb{S}} \Sigma_i)/k + \alpha$, where $k$ and $\alpha$ are two hyper-parameters, and $|\mathbb{S}| = k$. Here, $\mathbb{S}$ is the top-$k$ similar classes set to the novel sample $z$. $k$ denotes the number of base classes used to calibrate the novel class, and $\alpha$ controls the the degree of dispersion from the calibrated covariance. The augmented samples are then generated from $\mathcal{N}(\mu', \Sigma')$. In their experiments, they find calibrating from two base classes, *i.e.*, $k = 2$ achieve the best results. In our experiments, we set $k = 1, \alpha = 0$, since the number of base classes in our problem is 7, which is significantly smaller than 64, 160, and 100 base classes in natural images (Yang et al., 2021).

### D.4 ABLATION ON COVARIANCE TYPES

By default, LA estimates the "full" covariance matrix for each cluster. Following notations in GaussianMixture model API in scikit-learn library (Pedregosa et al., 2011), we further compute different types of covariance: 1) "tied", where all clusters share the same general covariance matrix estimated from the entire base dataset; 2) "diag", where each cluster has its own diagonal covariance matrix; 3) "spherical", where each cluster has its own single scalar variance. Specifically, "diag" covariance matrix only computes the variance of each channel (each dimension) without considering the correlation between channels, and "spherical" covariance further averages diagonal elements in each "diag" covariance matrix, resulting in a single scalar variance. Results are shown in Table 2.

Table 2 reports results of our default setting, where 16 clusters are used. Here we further explore more options of cluster numbers (leading to larger or smaller clusters). Figure D.2 show the results. LA with full covariance matrix does degenerate performance as the cluster size decreases (larger number of prototypes). In contrast, LAs with isotropic covariance matrices ("spherical"/"diag") perform more stably when cluster size alters. However, they underperform "full" covariance in all cluster sizes. Besides, similar performance degeneration is also observed when using isotropic covariance, but becomes slight. Note, the "Tied" covariance is estimated from the whole dataset.

Therefore the results should only be affected by randomness in latent augmentation (yellow line in Figure D.2).

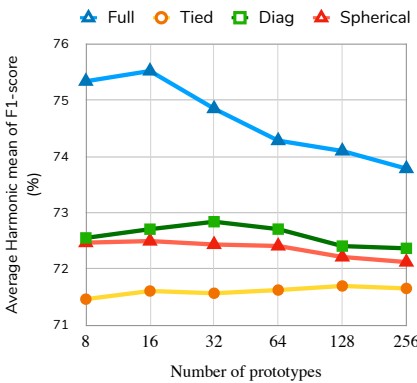

Figure D.2: Ablation study on cluster sizes v.s. covariance types.

## D.5 ABLATION ON K-MEANS RANDOM SEEDS

LA relies on K-Means clustering, which might be effected by random initialization. Here we explore how LA performs under different random seeds. Still, we use the K-Means API from faiss Johnson et al. (2019), a library for clustering. Table D.3 reports the results. LA is robust to the choice of random seeds. Among different seeds, the numerical differences of performance are within 95% confidence interval. We choose the random seed 66 for reproducibility in main paper.

Table D.3: Ablation on different K-Means seeds. Results of ablation tasks are reported, *i.e.* 300 meta-tasks under 5-shot setting.

| Random Seed | Base | Novel | HMean |
|---|---|---|---|
| Baseline | 85.85±0.78 | 53.27±1.63 | 65.74±1.06 |
| 10 | 86.85±0.84 | 65.70±1.40 | 74.81±1.05 |
| 20 | 87.34±0.80 | 66.74±1.33 | 75.66±1.00 |
| 30 | 87.71±0.81 | 66.48±1.35 | 75.63±1.01 |
| 40 | 87.31±0.80 | 65.04±1.39 | 74.55±1.02 |
| 50 | 87.83±0.78 | 66.40±1.35 | 75.63±0.99 |
| 66 (default) | 87.37±0.81 | 66.51±1.39 | 75.53±1.03 |
| Average | 87.40±0.81 | 66.14±1.37 | 75.30±1.01 |

## D.6 ABLATION ON COVARIANCE & L2 NORMALIZATION.

In our experiments, we compute the covariance matrices after $l_2$-normalization. To inspect whether the covariance is still meaningful after normalization. We compute the covariance matrices of NCT test set features before and after $l_2$-normalization. Then, we plot the covariance matrix as heatmaps. Since the length of a feature vector is 512, and inspecting a $512 \times 512$ heatmap might be difficult, we first compute the whole covariance matrix, then randomly select 128 channels to visualize; the upper triangle elements of this $128 \times 128$ matrix are masked out. Figure D.3, D.4, D.5 are results of three runs. We observe that the covariance before and after $l_2$-normalization are highly correlated, where the Pearson correlation coefficients are 0.9908, 0.9900, 0.9910 between before/after $l_2$-normalization covariance in Figure D.3, D.4, and D.5 respectively. Due to this high correlation, the covariance after $l2$-normalization may still be informative to some extend. Besides, a recent work on self-supervised learning, BarlowTwins (Zbontar et al., 2021), also compute covariance after $l_2$-normalization in their ablation study (Section 4, "Loss Function Ablations").

After $l_2$-normalization, all original samples are on the surface of a unit-hypersphere (a $d$-1 dimensional manifold, *i.e.* $\mathcal{S}^{d-1}$), but the augmented samples generated by LA are not necessarily on the surface. We therefore further normalize the augmented samples to see whether this impacts

performance. In the same setting as ablation study (300 meta-tasks, 5-sot), we observe the HMean marginally drops from 75.30±1.01 to 74.69±1.01. Although the $l_2$-normalized features are on the surface of a unit-hypersphere, the linear classifiers, typically, are still fitted in the original $\mathbb{R}^d$ space and no special consideration is taken. Generating augmented samples that are not on the surface of unit-hypersphere may still help. For more analysis about the niceness of unit-hypersphere, we refer the reader to the discussion section in Wang & Isola (2020).

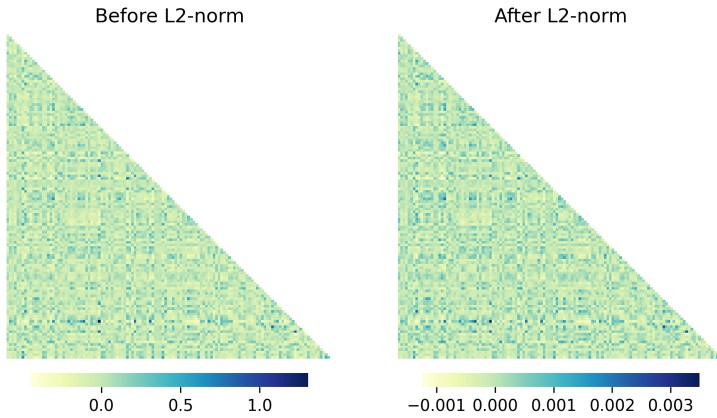

Figure D.3: **Heatmap visualization of the covariance matrix of randomly selected 128 channels before and after $l_2$-normalization.** Pearson correlation coefficients between them are 0.9908.

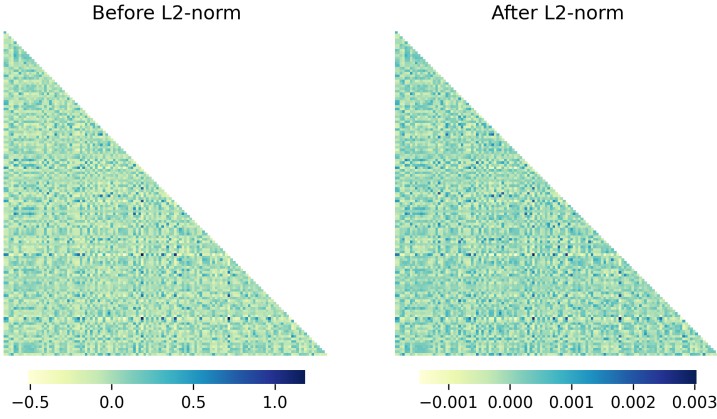

Figure D.4: **Heatmap visualization of the covariance matrix of randomly selected 128 channels before and after $l_2$-normalization.** Pearson correlation coefficients between them are 0.9901.

## D.7 COMPARISON TO DELTA-ENCODER

### D.7.1 REPRODUCTION

We follow an open-source repository[2] to implement $\delta$-encoder (Schwartz et al., 2018), and change the batch size from 128 to 512 for faster training speed. Other hyper-parameters remain unaltered.

Note that, $\delta$-encoder trains different generators for different shot settings, *e.g.,* one generator for 1-shot setting, and another for 5-shot. This differs from our flexible and universal augmentation pipeline. Besides, in both the official and the pytorch-version repositories, the best generator is chosen based on the best performance on **test set**. We do not follow this setting, and directly use the last-epoch model for augmentation.

---

[2]https://github.com/leven03/DeltaEncoder_pytorch

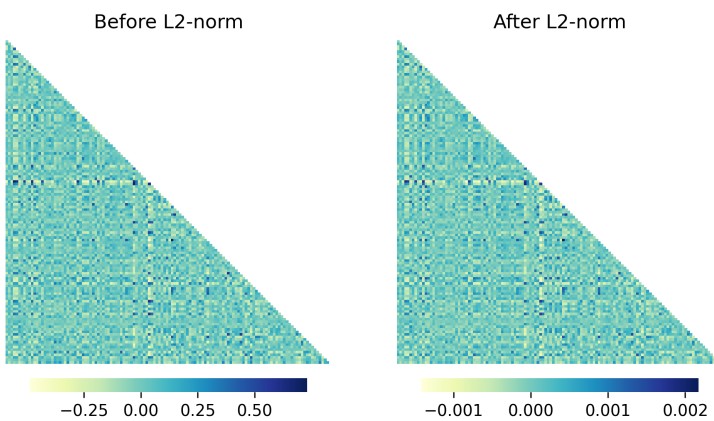

Figure D.5: **Heatmap visualization of the covariance matrix of randomly selected 128 channels before and after $l_2$-normalization.** Pearson correlation coefficients between them are 0.9910.

For 5-shot ablation task, we exclude the tumor-related classes to be novel class and use the remaining as base dataset to train $\delta$-encoder. All $\delta$-encoders are trained for 20 epochs (the original paper trains 12-epoch for 5-shot). We find the 20-epoch $\delta$-encoder performs slightly better than 12-epoch and 50-epoch $\delta$-encoders.

### D.7.2 RESULTS AND DISCUSSION

**Results.** Table D.4 shows the results. When using cluster labels as supervision, we find training $\delta$-encoder on non-$l_2$-normalized features and transferring to $l_2$-normalized features gives the best results. Normalizing the original features and generated features further improves the results ($63.40\% \rightarrow 64.92\%$). Using ground truth labels, unfortunately and surprisingly, leads to worse performance ($64.92\% \rightarrow 63.30\%$, especially no further $l_2$-normalization is applied on generated features ($63.30\% \rightarrow 48.11\%$). In our re-implementation, none of tested cases can outperform baseline and our method.

**Discussion.** During augmentation, $\delta$-encoder randomly samples a pair of base features from a same class, referred to as reference pair, and use it as guidance to augment novel samples. It, by construction, does not utilize any similarity information between base features and novel features. This could potentially cause problems as the base samples and novel samples can differ drastically, making the semantic transferring meaningless. In contrast, our LA queries the most likely variation from base dictionary via cosine similarity, which is a more strict constraint than randomly sampling as in $\delta$-encoder. Besides, in $\delta$-encoder, the reference pair used for augmentation could come from different base classes every time for a same novel class. For example, it could sample a LYM-LYM pair to augment one novel sample from TUM class, and use a ADI-ADI pair to augment another novel sample from TUM class[3]. This step might lead to feature inconsistency, which could degenerate performance.

### D.8 VISUALIZATION

### D.8.1 VISUALIZATION PROCEDURE

**Models.** All images are visualized with models that have never seen their classes during pre-training. For example, if the class "NORM" in NCT is regarded as novel class, samples belonging to it are excluded from this sub- pre-training dataset. Models trained on this sub-dataset are used to visualize "NORM" class.

**Similarity map.** To better understand why the generalization gap exists between CLP and FSP models, we visualize how CLP models and FSP models are attended to different local features for non-iconic, multi-object and multi-texture histology images. To this end, we resize images in

---

[3]LYM, ADI and TUM are two class names in NCT dataset

Table D.4: **Comparison to $\delta$-encoder in ablation task.** $\delta$-encoder is a supervised method. "Label source" column marks using unsupervised clustering labels or ground truth labels as supervision. We study training $\delta$-encoder on $l_2$-normalized (L2) and non-normalized (Non-L2) base features, marked in "Trained on" columns. Given the trained $\delta$-encoder, we study how the target features it is applied on affect the results, marked in the "Transfer to" column. At last, we study how $l_2$-normalization classifier affects final performance, marked in "Clf train&infer" column. Underlined numbers denote the best entries among each group rows. Bold numbers denote the best entries among all rows. Results are shown in percentage (%).

| Method | Label source | Train on | Transfer to | Clf train infer | Base | Novel | Hmean |
|---|---|---|---|---|---|---|---|
| $\delta$-encoder | KMeans(K=16) | L2 | L2 | L2 | 82.40 | 48.76 | 61.26 |
| | KMeans(K=16) | L2 | L2 | Non-L2 | 82.04 | 48.15 | 60.28 |
| | KMeans(K=16) | L2 | Non-L2 | L2 | 81.16 | 45.31 | 58.15 |
| | KMeans(K=16) | L2 | Non-L2 | Non-L2 | 77.89 | 45.58 | 57.51 |
| | KMeans(K=16) | Non-L2 | L2 | L2 | 79.68 | 54.77 | 64.92 |
| | KMeans(K=16) | Non-L2 | L2 | Non-L2 | 81.65 | 51.82 | 63.40 |
| | KMeans(K=16) | Non-L2 | Non-L2 | L2 | 81.52 | 48.21 | 60.59 |
| | KMeans(K=16) | Non-L2 | Non-L2 | Non-L2 | 77.82 | 46.89 | 58.52 |
| | Ground Truth | Non-L2 | L2 | L2 | 80.18 | 52.29 | 63.30 |
| | Ground Truth | Non-L2 | L2 | Non-L2 | 82.67 | 33.93 | 48.11 |
| ours | Baseline | | | | 85.85 | 53.27 | 65.74 |
| | Baseline+LA | | | | **87.68** | **66.71** | **75.77** |

the NCT test set to $448 \times 448$, and use the pre-trained CLP models and FSP models to extract $l_2$-normalized features from stage-2, stage-3, and stage-4 of ResNet-18. To obtain similarity heat maps, we compute the cosine similarity between global representations (after pooling) and local representations (before pooling). For absolute similarity ("Abs. Sim."), we directly rescale the similarity values by multiplying 255. For normalized relative similarity ("Rel. Sim."), we first rescale the similarity values to $[0, 1]$ by subtracting the minimum and dividing the maximum, and then multiply them by 255.

**Local feature agglomeration.** We follow Chen & Li (2020) to see how local features are agglomerated across layers. To this end, we run K-means in scikit-learn Pedregosa et al. (2011) on the $l_2$ normalized local features (before pooling) from stage-2, stage-3, and stage-4 of ResNet-18 with different numbers of clusters, i.e. 2, 4, 6.

### D.8.2   MORE EXAMPLES

Figure D.6 show more visualizations of a base class that has been exposed during both supervised pre-training and self-supervised pre-training.

Figure D.7 and Figure D.8 show more example images in the NCT dataset. Figure D.9 and Figure D.10 show the results of the LC-25K dataset and the PAIP dataset respectively. We visualize the absolute and relative cosine similarity between the global average pooled feature and the local features before pooling ("Abs. Sim." and "Rel. Sim." columns accordingly). To inspect correlation between local features, we conduct k-means clustering with different k values (columns with "k=*"). "Low", "middle" and "high" represent using features from stage-2, 3, and 4 from ResNet-18.

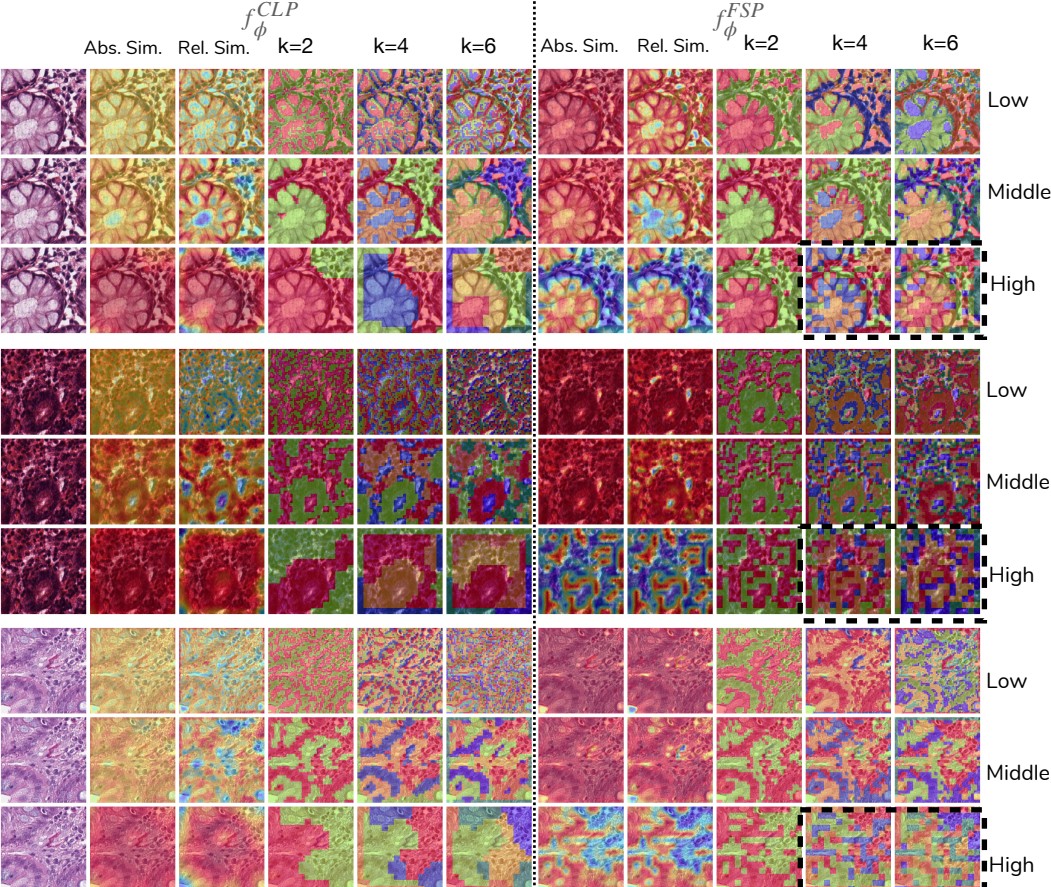

Figure D.6: **Visualization of base classes in NCT dataset.** FSP show high correspondence to the most discriminative parts (Abs. Sim. and Rel. Sim. columns). However, the cluster results of non-discriminative parts appear to be meaningless, which may implies FSP's failure in encoding all available semantic information in the image.

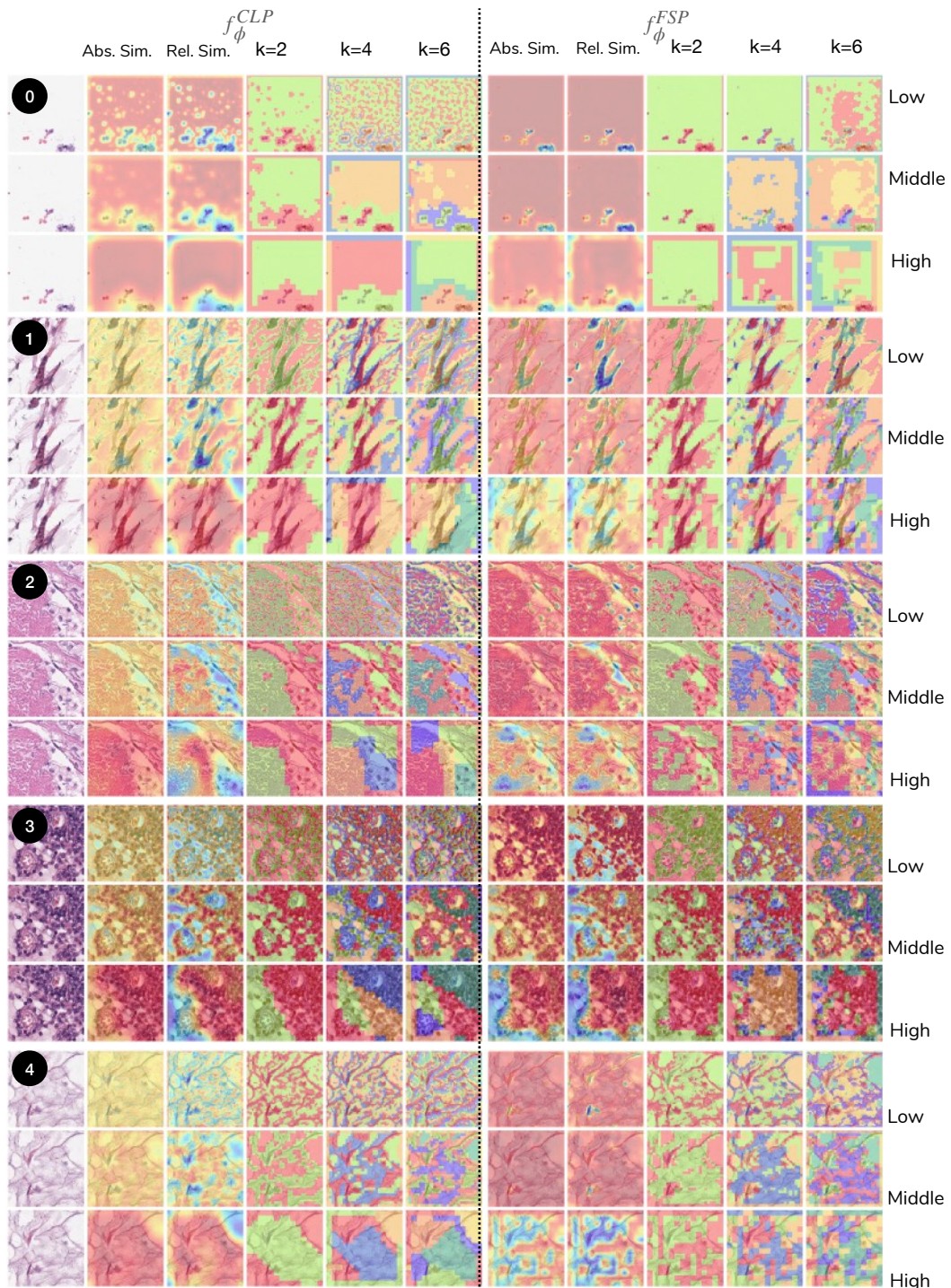

Figure D.7: **Visualization in the NCT dataset - 1.** White indexes in black circles represent class IDs. They are: 0) background (BACK), 1) adipose (ADI), 2) debris (DEB), 3) lymphocytes (LYM), and 4) mucus (MUC).

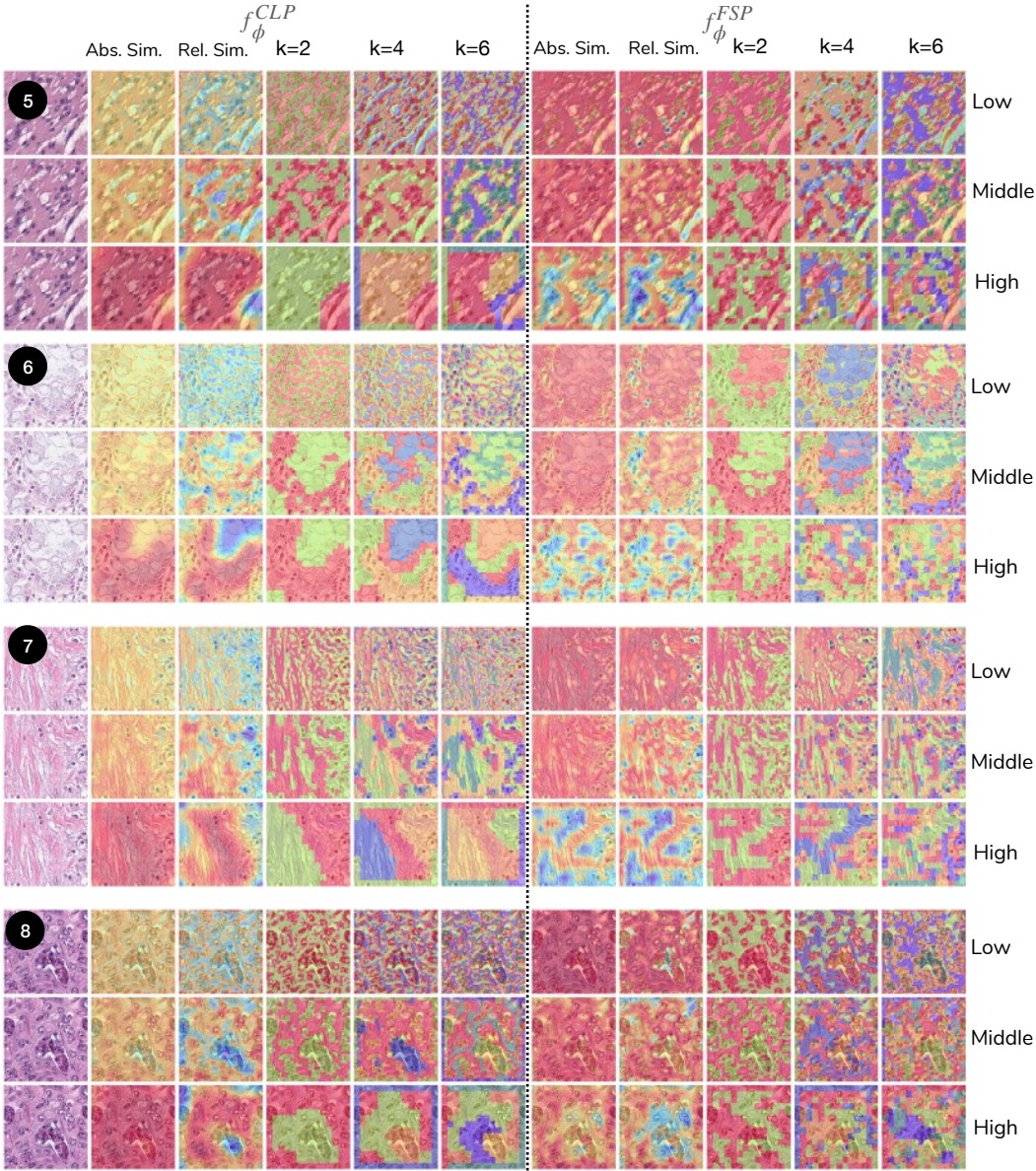

Figure D.8: **Visualization in the NCT dataset - 2.** White indexes in black circles represent class IDs. They are: 5) smooth muscle (MUS), 6) normal colon mucosa (NORM), 7) cancer-associated stroma (STR), 8) colorectal adenocarcinoma epithelium (TUM).

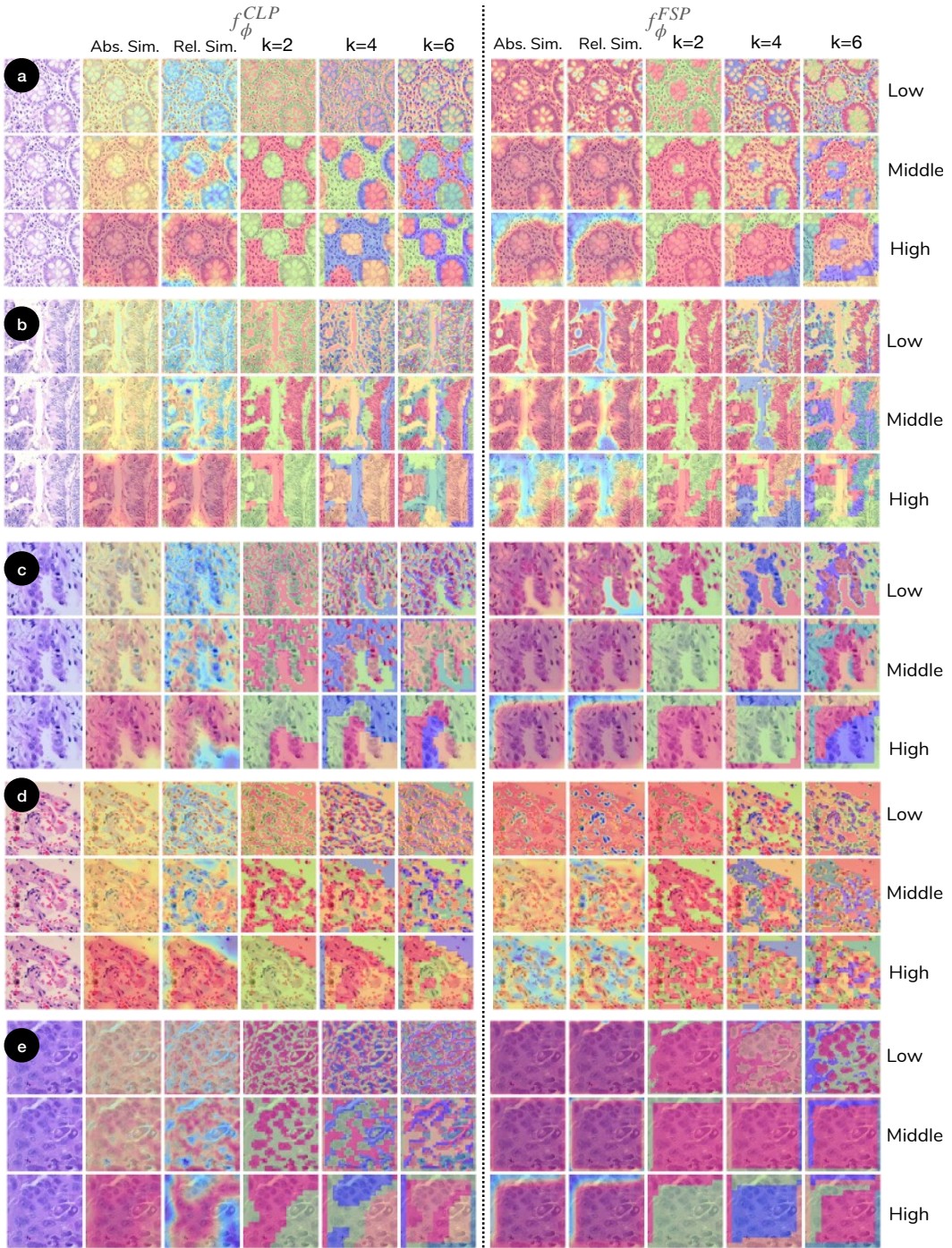

Figure D.9: **Visualization in the LC-25K dataset.** White indexes in black circles represent class IDs. They are: 0) colon adenocarcinoma, 1) benign colonic tissue, 2) lung adenocarcinoma, 3) benign lung tissue, and 4) lung squamous cell carcinoma.

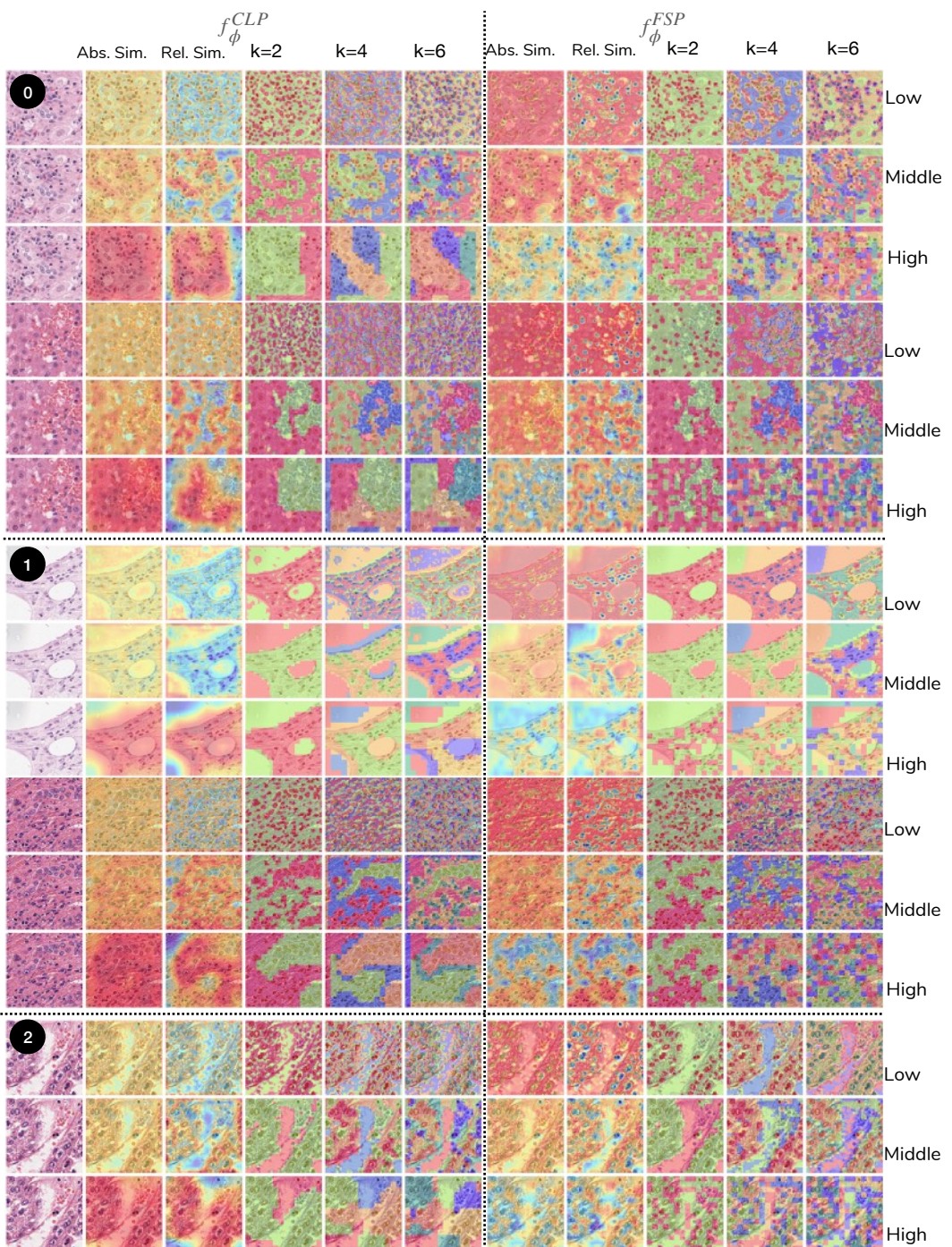

Figure D.10: **Visualization in the PAIP dataset.** White indexes in black circles represent class IDs. They are: 0) non-tumor, 1) viable-tumor and 2) other tissues.

