# OpenReview forum: "Towards Better Understanding and Better Generalization of Low-shot Classification in Histology Images with Contrastive Learning"
_ICLR.cc/2022/Conference — ICLR 2022 Poster_

### Official Review · Reviewer_BmyB · 2021-10-26

**Correctness:** 3
**Technical Novelty And Significance:** 3
**Empirical Novelty And Significance:** 3
**Recommendation:** 6
**Confidence:** 3

**Main Review:**

The topic of few-shot learning in biomedical applications is of great interest, in particular in histology where whole slide scanners can capture a lot of data, but annotations are scarce and expensive. The datasets used in the study are openly available and cover a large range of variations in tissue, staining and pathologies.

The paper is well structured, but hard to follow. The language/text of the paper should be revised. At various places expressions used are ambiguous which affects the scientific value to the point of incorrectness and makes the steps taken very hard to understand. Articles are often omitted, there are a number of grammatical mistakes and some words are used erroneously throughout the paper (reflect, confront, deprecate...).

Some vocabulary used is not conventional or precise. Some examples:
These two components [...] can scale gracefully [...]. (abstract)
[...] CLP generalizes fairly better [...]. (sec. 4.2)
Such boost [...] becomes humble in out-domain task. (sec. 4.2)
[...] DA can slightly polish LA [...]. (sec 4.3)

It should be addressed more clearly what is meant by “near-domain”, “mixture-domain”, “middle-domain” and “out-domain”.
In 4.1 it says “We use faiss to perform K-means with a fix seed.” - it should be pointed out that faiss is a library for clustering.
For readers outside this immediate field of deep learning (e.g. more application oriented bioimage analysts in digital pathology), I suggest to use the same terminology throughout the paper, currently the authors switch for example between “low-shot” and “few-shot” and similar.

At multiple places it was hard to understand what the authors were trying to say, a few examples are:
in 3.2.1 “From the view of low-data learning, distribution of few samples is not well calibrated, so using established distributions in base class to calibrate untrustworthy novel class can help.”;
in 4.1 “The rest three lung-related classes are regarded as out-domain novel classes.”;
in 4.3 “It is worthy to emphasize the computation budget involved in LA (addition in $\mathbb{R}^d$ space) is significantly lower than LA (image augmentation in $\mathbb{R}^{3HW}$ space and encoder forwards)“
In 4.3 “Patches from a single WSI are confronted to its stain degree (impacts color) and body site (impacts pattern and context)” - what does it mean they are confronted to these properties?
In 4.4 “The lack of dominant objectiveness makes FSP tend to overfit severely in high level, but CLP can capture the overall patterns from the crudely scattered tissues.”


Scientific content:
FSL description in Sec. 2: Is $\mathcal{S}$ a subset of $\mathcal{D}$_{novel} and $\mathcal{Q}$ a subset of $\mathcal{X}$_{novel}? If so, the notation needs to be revised. Currently it states they are distributed by $\mathcal{D}$_{novel} and $\mathcal{X}$_{novel} respectively. Furthermore, the sets $\mathcal{S}$ and $\mathcal{Q}$ are indexed by i=1 to NK and i=1 to NQ, but immediately after K and Q are referred to what probably should be NK and NQ.
The authors investigate how augmentation in the latent space compares to data augmentation w.r.t. To F1-score. However, I cannot find the specifications of what kind of data augmentation was used in the main paper, which is a crucial aspect. I eventually found some details in the appendix, but I would like to see some intuition behind the choice of data augmentation (crops, horizontal flips and color jittering) and how it is expected to be comparable to the latent augmentation in terms of adding variety to the dataset. It is common to use more extensive data augmentation - rotations, mirroring, blurr,... I also lack the information if at every iteration one augmentation was drawn at random, or a combination of the augmentations.
Additionally, the data augmentation used for the fully supervised pretraining (crops, horizontal flips and normalization) was chosen differently than in the ablation study, why?
It is not clear to me what is meant with augmentation times in the experiments of LA vs DA. Is it that the augmentations are being performed on already augmented versions of the original input? If so, how does the additive color jittering affect the images?
How are the patches resized in the LC-25K dataset? What interpolation is used?
Fig. 4, visualizations of CLP features from stage 4 using 4 and 6 clusters for k-means: it looks as if there are three clusters inherent to the data present in the shown image, but for k=4 and k=6, the data is split into 4 or 6 clusters due to how k-means work, but I wonder why those “additional” clusters are distributed along the edges of the image as they are? Are the edge effects in these feature maps that cause this behavior or is it due to initialization of the seeds for k-means? Similar behavior can be observed in the visualizations provided in the appendix.

Novelty:
The authors say that few-shot learning “is much unexplored in medical images, especially in histological ones.” and they “pioneer the study of low-shot learning for histology images [...]”. However, there are a number of studies on this topic, some included in the paper’s related work section, but there are other recent works on this topic also, “Learning with Less Data Via Weakly Labeled Patch Classification in Digital Pathology“ by Teh et al. (ISBI 2020), “Supervision and Source Domain Impact on Representation Learning: A Histopathology Case Study” by Sikaroudi et al. (EMBC 2020) to name a few.


**Summary Of The Paper:**

The paper addresses few-shot classification for histological images. They compare the generalization capabilities of self-supervised contrastive learning and fully supervised classification as pretraining. They perform experiments to evaluate the impact of latent augmentation vs. data augmentation. They use three publicly available histological datasets (spanning multiple tissues, pathologies and number of available classes). They construct a number of tasks given the tissue labels provided in those datasets and compare the classification performance in N-way-K-shot classification experiments for a number of different N and K.


**Summary Of The Review:**

The paper addresses few-shot learning for the classification of histological images. This topic is of significant interest to the biomedical community, as annotations are very expensive to obtain. Exploring the generalizability of different pretraining methods and augmentations to exploit the available ground truth to its fullest extent is of great importance. However, I found it very hard to understand the text and followingly some of the experimental setup. On multiple occasions, the phrasing was so ambiguous, that it became unclear to me what the authors tried to convey. The text needs revisions to correct for semantic and grammatical errors. Due to this unclarity, I suggest to reject this submission, but encourage the authors to revise the text and submit to another venue.



------------- After author's response and revision of the submission ------------------
The authors addressed my concerns and questions in detail and the revised submission has improved in clarity. I believe the topic is of significant interest to the biomedical imaging community, which is why I am suggesting to accept the revised submission.

---

> ### Author Response · Authors · 2021-11-19
> **Response to Reviewer BmyB (1/3)**
>
> We thank Reviewer BmyB for the constructive comments and some positive feedback on our work.
>
> Below we address each concern separately.
>
> > Q1. Writtings and typos.
>
> We apologize for any misunderstanding caused by writing. We have revised our paper and promise a more thorough examination in the camera-ready version.
>
> > Q2. Some specific terms.
>  * "Near-/mixture-/middle-/out-domain": When novel and base classes are from different organs, we view the novel classes as out-domain classes. When they are from the same organ, we consider them as near-domain classes if the data is collected from the same source under similar protocols. Otherwise, they are considered as middle-domain classes due to the difference in imaging protocols. In our paper, samples in the NCT dataset are considered to be in the near-domain (they are from the colon and are collected under similar protocols). Two colon-related classes in LC-25K dataset are considered as middle-domain classes (they are from the colon site but are collected under a different protocol compared to NCT's). The rest three lung-related classes in LC-25K dataset are considered as out-domain classes (a different organ). All classes from the PAIP dataset are out-domain (a different organ, lung). Mixture-domain: since LC-25K has middle-domain classes and out-domain classes, we consider the recognition tasks on it as mixture-domain tasks. We have revised the paper accordingly to clarify this.
> * "low-/few-shot": we have revised to use "few-shot" consistently along with the text. Thanks for the suggestion.
> * "From the view of low-data learning, distribution of few samples is not well calibrated": this sentence is to summarize the related work [1]. An in-depth discussion is out of the main scope of this paper. We suggest referring to [1] for more details.
> * "Computation budget": sorry for the typo. It should be "the computation budget involved in LA (addition in $\mathbb{R}^d$ space) is significantly lower than **DA** (image augmentation in $\mathbb{R}^{3HW}$ space and encoder forwards)". Typically, a data augmentation requires image-level computation in RGB space (e.g. 3x224x224, or $R^{3HW}$). After augmentation, each augmented image should be fed into the encoder to extract features (encoder forwards), resulting in many times forwards. In contrast, LA only requires one-time forward, followed by querying the most similar prototypes in "legacy dictionary" (now revised to "base dictionary"), and augmenting samples via addition operation, i.e. $\hat{z} = z + \delta$, where $\delta$ is sampled from the multivariate Gaussian distribution and $z, \delta \in \mathbb{R}^d$. We pre-sampled enough shift vectors $\delta$ for each prototype (2000 shift vectors for each) so the sampling process does not cost time. Addition in $R^{d}$ space is significantly cheaper than augmentation in $\mathbb{R}^{3HW}$ space and encoder feedforwards.
> * "The lack of dominant objectiveness makes FSP tend to overfit severely in high level, but CLP can capture the overall patterns from the crudely scattered tissues": We have removed this sentence and added more discussion there (Why do CLP models generalize better than FSP ones in histology images? in Section 4.4).
>
>
> ### Scientific content
> > Q3: Is S a subset of D{novel} and Q a subset of X{novel}? If so, the notation needs to be revised. Currently, it states they are distributed by D{novel} and X{novel} respectively. Furthermore, the sets S and Q are indexed by i=1 to NK and i=1 to NQ, but immediately after K and Q are referred to what probably should be NK and NQ.
>
> **A3:** Yes, $\mathcal{S}$ and $\mathcal{Q}$ are two very small subsets of $\mathcal{D}${novel/joint} and $\mathcal{X}${novel/joint} respectively. By using symbol "$\sim$", we mean $\mathcal{S}$ and $\mathcal{Q}$ are **sampled** from $\mathcal{D}${novel/joint} and $\mathcal{X}${novel/joint} respectively. Every meta-testing task is constructed by different support sets and query sets.
>
> Regarding the index, a typical N-way K-shot Q-query task is that: a support set contains K samples for each of N classes. This gives us NK samples for training. Similarly, a query set contains Q samples for each of N classes, giving a total of NQ samples for testing. We follow the conventional problem formulation in few-shot classification.

---

> > ### Author Response · Authors · 2021-11-19
> > **Response to Reviewer BmyB (2/3)**
> >
> > > Q4.1: T The authors investigate how augmentation in the latent space compares to data augmentation w.r.t. To F1-score. However, I cannot find the specifications of what kind of data augmentation was used in the main paper, which is a crucial aspect. I eventually found some details in the appendix,
> >
> > **A4.1:** Thanks for noting the missing reference. We have specified an explicit reference there in the revised paper and modify text in Appendix D.2 to: "The data augmentations used in this experiment are RandomResizedCrop(scale=(0.8, 1.0)) with probability 1.0, RandomHorizontalFlip with probability 0.5, and ColorJitter(brightness=0.4, contrast=0.4, saturation=0.4, hue=0.2) with probability 0.8. These augmentations are applied sequentially and jointly on original images."
> >
> > > Q4.2: but I would like to see some intuition behind the choice of data augmentation (crops, horizontal flips, and color jittering) and how it is expected to be comparable to the latent augmentation in terms of adding variety to the dataset. It is common to use more extensive data augmentation - rotations, mirroring, blurr,...
> >
> > **A4.2:** We choose these data augmentations as they are fast to compute.
> > Color jittering can simulate different stainings to some extent. Random cropping, flipping are two common and fast data augmentations. Currently, we have not included the experiments using more data augmentation, e.g. rotations, mirroring, blur, due to the high computational cost and limited time. To get a sweet spot from data augmentation, according to Figure-3(b), a 49 ~ 64 times more computation budget is needed. This brings significantly more computation budget but, likely, no better results than LA.
> >
> > And we would like to emphasize that the advantage of LA in the few-shot problem is not only the improved accuracy but also the significantly reduced computational complexity, i.e. DA requires augmentation in $R^{3HW}$ space and an encoder forward each time. When augmenting the original images 100 times, the time budget would be significant. In contrast, LA only requires querying and adding operation in $R^{d}$ space. 100-times LA only costs 1 querying and 100 adding operations.
> >
> > > Q4.3: I also lack the information if at every iteration one augmentation was drawn at random, or a combination of the augmentations.
> >
> > **A4.3:** The data augmentations are applied sequentially in the order stated in our paper, i.e. a combination of augmentations are applied sequentially and randomly. The data augmentations are RandomResizedCrop(scale=(0.8,1.0)) with **p=1.0**, RandomHorizontalFlip with **p=0.5**, and ColorJitter(brightness=0.4, contrast=0.4, saturation=0.4, hue=0.2) with **p=0.8**. We have clarified them in the appendix of revised paper.
> >
> >
> > >Q4.4: Additionally, the data augmentation used for the fully supervised pretraining (crops, horizontal flips, and normalization) was chosen differently than in the ablation study, why?
> >
> > **A4.4:** One of our primary goals is to study the generalization gap between FSP and CLP in our problem. We would like to show the large generalization gap in our experiment is larger than that in the ImageNet dataset. Therefore, we directly follow the fully supervised pre-training setting in OpenSelfSup [2], a self-supervised-learning-oriented codebase, to conduct our experiments. In their setting, only crops, horizontal flips, and normalization are used. We hope this would demonstrate the generalization gap is mainly attributed to the datasets difference or the difference between natural images and histological images.
> >
> > Besides, Chen et. al.[3] have explored the effects of longer training and data augmentation for supervised pre-training in their SimCLR paper[3] (Appendix B.3.) that they observe no significant benefit from training supervised models longer and with stronger data augmentation.
> >
> >
> > >Q4.5: It is not clear to me what is meant with augmentation times in the experiments of LA vs DA. Is it that the augmentations are being performed on already augmented versions of the original input?
> > If so, how does the additive color jittering affect the images?
> >
> > **A4.5:** No, all augmentations are applied on the original inputs, sequentially and randomly.
> >
> > > Q4.6: How are the patches resized in the LC-25K dataset? What interpolation is used?
> >
> > **A4.6:** They are resized to 224x224 by bilinear interpolation.

---

> > > ### Author Response · Authors · 2021-11-19
> > > **Response to Reviewer BmyB (3/3)**
> > >
> > > > Q5: Fig. 4, visualizations of CLP features from stage 4 using 4 and 6 clusters for K-Means: it looks as if there are three clusters inherent to the data present in the shown image, but for k=4 and k=6, the data is split into 4 or 6 clusters due to how k-means work, but I wonder why those “additional” clusters are distributed along the edges of the image as they are?
> > > Are the edge effects in these feature maps that cause this behavior or is it due to initialization of the seeds for k-means? Similar behavior can be observed in the visualizations provided in the appendix.
> > >
> > > **A5:** We also notice this behavior, and we conjecture that the zero-padding operation in ResNet inevitably convolutes zeros into feature maps, resulting in an "edge effect". Therefore, K-Means may group features close to the same border to the same cluster. This can also be observed in Chen & Li (2020) [4].
> > >
> > >
> > > > Q6: Related work.
> > >
> > > **A6:** Thanks for pointing out some related works. We have added a discussion to them in the revised paper.
> > >
> > > ---------------
> > > ### Reference
> > > [1] Yang, Shuo, Lu Liu, and Min Xu. "Free lunch for few-shot learning: Distribution calibration." arXiv preprint arXiv:2101.06395 (2021).https://arxiv.org/pdf/2101.06395.pdf
> > >
> > > [2] OpenSelfSup, https://github.com/open-mmlab/OpenSelfSup
> > >
> > > [3] Chen, Ting, et al. "A simple framework for contrastive learning of visual representations." International conference on machine learning. PMLR, 2020. http://proceedings.mlr.press/v119/chen20j/chen20j.pdf
> > >
> > > [4] Chen, Ting, and Lala Li. "Intriguing properties of contrastive losses." arXiv preprint arXiv:2011.02803 (2020), Chen & Li (2020): https://contrastive-learning.github.io/intriguing

---

> > > > ### Comment · Reviewer_BmyB · 2021-11-21
> > > > **Addressing Author's Response**
> > > >
> > > > I thank the authors for addressing my concerns and questions in such detail, I appreciate the effort to clarify these points. I do believe the revised submission has improved in clarity.
> > > >
> > > >
> > > > Some comments on the revised submission:
> > > >
> > > > "Two of five classes in LC-25K are colon-related (middle-domain novel classes) and the rest three are lung-related (out-domain novel classes)" should be "Two of five classes in LC-25K are colon-related (middle-domain novel classes) and the remaining three are lung-related (out-domain novel classes)"
> > > >
> > > > "K-means with a fix seed for reproducibility" should be " K-means with a fixed seed for reproducibility".
> > > >
> > > > ImageNet is misspelled as "ImgeNet" at the bottom of page 8.
> > > >
> > > > I appreciate the effort to settle on "few shot" instead of switching between "few shot" and "low shot" throughout the paper - I noticed that Fig 2 still refers to "low-shot classification".

---

> > > > > ### Author Response · Authors · 2021-11-21
> > > > > **Thank you!**
> > > > >
> > > > > Thanks for your kind correction! We have revised the text and figure accordingly. We sincerely appreciate your time and any suggestion on our work.

---

### Official Review · Reviewer_qUzb · 2021-10-29

**Correctness:** 3
**Technical Novelty And Significance:** 2
**Empirical Novelty And Significance:** 3
**Recommendation:** 8
**Confidence:** 4

**Main Review:**

Main strengths:
1) The paper focuses on a real limitation which hinders widespread deployment of ML-based pipelines in histological image analysis: the issues relative to out-of-domain adaptation, especially with respect to the introduction of a new label set (i.e., different types of tissues/anomalies).
2) The proposed approach, which makes large use of some of the latest development in both FSL and data augmentation strategies, is interesting and convincing.
3) Most importantly, the experimental section is comprehensive and well-devised. Three different adaptation scenarios are implemented, with varying degrees of domain adaptation required.

Main weaknesses:
1) Lack of novelty: while probably novel in their combination and use case, none of the implemented techniques (i.e., contrastive and fully-supervised classification, domain adaptation through FSL) is very novel. The only exception might be “latent augmentation”, which follows in the footsteps of previous latent space augmentation techniques but - differently from most of them - doesn’t require labels. This aspect is interesting, but since the experiments are limited to histological images, it is difficult to assess its impact on image classification in general.
2) Potentially questionable take-home message: amongst their most important contributions, the authors state that “Also to our surprise, we show that state-of-the-art CL models generalize fairly better than the supervised counterparts for histological images”. In their Discussion, they add “The lack of dominant objectiveness makes FSP (Fully-supervised pre-training) tend to overfit severely in high level, but CLP (contrastive-learning pre-training) can capture the overall patterns from the crudely scattered tissues. We believe this is the main reason why CLP models generalize better than FSP models, in pathology.” To me, this is more a statement about the observed outcome rather than an explanation. Additionally, I’m not sure that the results shown by Chen & Li (2020) are directly comparable to these, which are relative to the same comparison but performed in an out-of-domain (i.e., previously unseen class) setting. It seems rather intuitive to me that FSP-learned representations would be unreliable when a patch showing a previously unseen class is presented. In CLP, on the other hand, the goal is learning to identifying similarities and differences in images, which should provide more class-agnostic mid-level features. I would appreciate a more detailed discussion of this comparison.

Minor comments:
1. Typos (page 5): “samples belong”, “we summarizes”
2. Typo (page 15): “oragan”. Also, a “plus” seems missing from the “Out” equation.
3. Typo (page 7): “The trade-off EXISTS”
4. Typos (page 9): “Later methods is attracted to transfer learning”, “This method is later extended by not relying given base samples”
5. Please consider citing and briefly discussing the following paper on latent space augmentation from ICLR 2021: Tsz-Him Cheung, Dit-Yan Yeung. MODALS: Modality-agnostic Automated Data Augmentation in the Latent Space. ICLR 2021

Chen & Li (2020): https://contrastive-learning.github.io/intriguing

**Summary Of The Paper:**

The paper focuses on learning useful representations of histological patches for classification tasks. Acknowledging the large domain shifts typical of histological imaging (e.g., differences in examined organs/tissues, imaging protocols, staining intensities), the authors frame the classification problem as a few-shot learning (FSL) one. In their experiments, they compare FSL pre-training on a base dataset both using full supervision and contrastive learning. During low-shot training, they make use of an unsupervised approach for data augmentation in the latent space (referred to as “latent augmentation”). In the results, they note that contrastive learning with latent augmentation provides the most reliable representations, especially for out-of-domain tasks.

**Summary Of The Review:**

Overall, I vote for acceptance. While the novelty might be relatively limited, the comparison between fully-supervised and contrastive pre-training in an FSL setting (with and without latent augmentations) for histopathological images is interesting. The experimental design and ablations provide a valuable overview of the inner workings of these methods, which could be valuable to a large portion of ICLR’s audience (potentially beyond applications to histopathology).

---

> ### Author Response · Authors · 2021-11-19
> **Response to Reviewer qUzb (1/3)**
>
> We sincerely thank Reviewer qUzb for the detailed and valuable comments and the recognition of our efforts.
>
> > Q1. Lack of novelty: while probably novel in their combination and use case, none of the implemented techniques (i.e., contrastive and fully-supervised classification, domain adaptation through FSL) is very novel. The only exception might be “latent augmentation”, which follows in the footsteps of previous latent space augmentation techniques but - differently from most of them - doesn’t require labels. This aspect is interesting, but since the experiments are limited to histological images, it is difficult to assess its impact on image classification in general.
>
> **A1**: We plan to move forward to see whether the unsupervised, or self-supervised to some extent, latent augmentation (LA) generalizes to natural images or other kinds of images such as satellite images. Besides LA's effectiveness, we would like to re-emphasize the intriguing large generalization gap between fully-supervised pre-training (FSP) and contrastive-learning pre-training (CLP) we observed in our experiments. We defer more discussion on it later to **A2**.
>
> This work does build upon the footsteps of previous meritorious works, e.g. MoCo-v3 and supervised variation methods that are clearly stated in our related work section, and we have not overclaimed its technical novelty. We show how few-shot classification in histology images can benefit from the combination of contrastive learning pre-training (CLP) and a self-supervised latent augmentation (LA) method, with decent gains. We notice the large generalization gap between fully-supervised pre-training (FSP) and CLP in our problem. Thereafter, we try to analyze what might contribute to better generalizability. That's why we entitle our work part as **"Towards better understanding and better generalization"**. Since few-shot learning in histology images is at its early stage, and self-supervised learning is a fast-growing field, we hope this work would provide useful data points and experience for them, from a representation learning perspective (more discussion about empirical findings and why they matter in **A2**.)

---

> > ### Author Response · Authors · 2021-11-19
> > **Response to Reviewer qUzb (2/3)**
> >
> > > Q2. Potentially questionable take-home message [...] To me, this is more a statement about the observed outcome rather than an explanation. I’m not sure the results shown by Chen & Li (2020) are comparable to those. It seems rather intuitive to me that FSP-learned representations would be unreliable when a patch showing a previously unseen class is presented. In CLP, on the other hand, the goal is learning to identify similarities and differences in images, which should provide more class-agnostic mid-level features. I would appreciate a more detailed discussion of this comparison.
> >
> > **A2:** We sincerely appreciate this comment. It triggers our deeper analysis and further thinking of why CLP generalizes better in our problem.
> >
> > *Why we say the visualization in our problem differs from Chen & Li (2020)*: they visualize how SimCLR (a CLP model) and a supervised model (an FSP model) group feature maps. There, FSP and CLP cluster features in a similar way: grouping features in a part-to-whole, element-to-object manner. But FSP and CLP act differently in our problem. In most cases, only CLP groups feature in a part-to-whole, element-to-object manner (Figure 4 in our paper). When the number of clusters increase (e.g. k=6 in our paper Figure 4), the disparity between FSP and CLP becomes larger, i.e. FSP's clusters become less semantic meaningful.
> >
> > > It seems rather intuitive to me that FSP-learned representations would be unreliable when a patch showing a previously unseen class is presented.
> >
> > We appreciate your sharing and discussion on this point! We have not yet come to this point before.
> >
> > In this rebuttal period, in addition to visualization on novel classes, we further visualize base classes (the bottom row of Fig. 4 and more in Fig. D.6.) and find the disparity between FSP and CLP also exists. In the bottom row of Figure 4 (visualization for a base class sample), FSP only pays attention to the most discriminative parts, leaving the rest “redundant parts” -- the parts contribute less to the recognition of this class -- disorder (dashed box in Fig. 4 and Fig. D.6). However, the most discriminative parts are likely to alter when a new class is presented. FSP’s inability to fully encode meaningful information may lead to its failure in generalizing to new classes. In other words, what FSP remembers or learns is the discriminative parts of the base classes. Meanwhile, CLP encodes most of the tissue-structure-related features without knowing which parts are more discriminative. This behavior may be useful for novel class recognition and may result in better generalizability.
> >
> > *How can this be? Or what findings and observations in our paper may trigger?:* We notice the visualization difference and the large generalization gap between CLP and FSP when pre-training in a histology image dataset, or more specifically, NCT dataset -- a dataset has 9 classes and 100k samples. ImageNet has more diverse classes (1000 classes) and samples (∼1.28M images) than those in histology datasets. FSP models in ImageNet need to recognize the discriminative parts of all 1000 classes. In such a case, the redundant information in one class might contribute to the recognition of another class. Therefore FSP may eventually encode most of the available information for new classes that are related to ImgeNet classes. However, histology datasets usually lack enough diverse annotated classes that help to build a know-everything FSP model. We should point out that our discussion there is about what the reasons behind this large gap can be. The large gap in our problem still remains as an empirical observation. We believe the observations leave room for future research on studying FSP and CLP, for example:
> > * Is it the label diversity of pre-training datasets that causes a larger generalization gap? FSP and CLP models are both pre-trained on a  9-class dataset (NCT dataset) in our problem. Previous works train FSP and CLP models on a 1000-class dataset (ImgNet dataset). The label diversity of ImgNet is significantly larger than NCT. Will this impact the FSP models' generalizability as we hypothesize at the end of the discussion section (in the revised paper)?
> > * Is the sample diversity that causes a larger generalization gap? histology patches usually have a similar appearance and have less diversity compared to natural images.
> > * Is it the dataset scale that causes a larger generalization gap? NCT dataset has 100k samples while ImgNet has ~ 1.28M samples.
> > * Is it the histology image itself that causes a larger generalization gap? histology images usually have repeated patterns and textures and lack dominant objects, which makes them different from natural images.
> >
> > We now modify and add more discussion on this topic in the revised paper in Section 4.4. (**Why do CLP models generalize better than FSP ones in histology images?**).
> >
> > This is overall an open question and needs further exploration.
> >
> > Again, we do appreciate your intriguing comments!

---

> > > ### Author Response · Authors · 2021-11-19
> > > **Response to Reviewer qUzb (3/3)**
> > >
> > > ### Minor comments
> > > > Typos
> > >
> > > Thanks for the correction! We have revised them accordingly.
> > >
> > > > Discussion on related work
> > >
> > > We have included the discussion in the related work section as well as the intuitions and motivations on latent augmentation sections (Sec. 3.2.1).
> > >
> > > Thanks for your detailed comments!

---

### Official Review · Reviewer_hJxe · 2021-11-01

**Correctness:** 4
**Technical Novelty And Significance:** 3
**Empirical Novelty And Significance:** 4
**Recommendation:** 8
**Confidence:** 4

**Details Of Ethics Concerns:**

-

**Main Review:**

**Strengths**
- Great results, showing a consistent and significant performance boost (~ +20% F1 score) few-shot classification of histology images.
- The proposed latent augmentation is simple to implement, and it consistently outperforms the standard input augmentation. It has the potential to be adopted by the community.
- Very well structured and written paper. The paper feels complete.
- Extensive experiments and ablation studies, with sufficient implementation details provided (e.g., MoCo method is explained)
- Extensive discussion of the findings.


**Weaknesses**
- The proposed method, "latent augmentation" isn't compared to other variation data augmentation methods. This makes the reader wonder whether "latent augmentation" is superior to other variation augmentation methods. Why didn't you compare with other " variation augmentation" methods? For example, how well does adding white noise to the latent variables perform?
- Add a small investigation of the shape of the clusters (i.e., the covariance matrix). K-Means corresponds to GMM with an isotropic covariance matrix. If it turns out that isotropic covariance matrices are sufficient, this information would be helpful in scenarios with small clusters (when the covariance approximations are poor).
- How stable is the method to the randomness in K-means? Did you notice any significant performance variation when running K-means with different random seeds?

**Minor comments**
- The experiment (and the caption) of figure 4 is unclear, even though the appendix provides many details. Please refactor the explanation so that it doesn't require reading the appendix (for example, explain what the features are inside the caption).
- It's unclear how l2 feature normalization is used in clustering and downstream meta-tasks. As of now, I understand that l2 feature normalization is applied before doing clustering. However, all feature vectors would be on the unit hypersphere, giving a meaningless cluster covariance matrix. Please clarify.
- Some paragraphs are too general and can be reduced in size. For example, the first paragraph in section 3.1 could be smaller. The fully supervised pre-training (equation 1) is just the standard cross-entropy loss. The few-shot learning section from related works could be reduced in size by a third.
- Replace the formula for choosing the cluster in latent augmentation (penultimate row in section 3.2) with simply saying that i* is selected by using cosine similarity between the representation z and c_i.
- The following terms are ambiguous and should be changed:
saying that the results "fairly better."
-- "space is shown to be highly linearized" is ambiguous. Do you mean the space allows linear interpolation?
-- "legacy dictionary"
- Does the +- numbers in Table 1 specify the confidence interval or the standard deviation across multiple runs?
- Replace "golden labels" with "true labels"
- Typo: "significantly lower than LA (image augmentations" should be DA instead of LA
- Will the code be released?


**Summary Of The Paper:**

The paper investigates few-shot learning for histology images. So far, contrastive learning has been studied for natural images, which differ from medical images. The paper shows that contrastive pre-training provides a significant performance boost (~ +10% F1 score) compared to standard supervised pre-training in few-shot learning. Further, they introduce a new self-supervised method (called "latent augmentation") for performing augmentation in the feature space created by the model. The prediction model (e.g., logistic regression) trained on the augmented data provides ~ +10% higher F1 score for small and moderate shifts in the pre-training and base dataset (for pre-training the feature extractor) and the support dataset (in the meta-tasks). Overall, the results are very promising, and the performance boost is significant. The paper has the potential to make contrastive learning the standard pre-training methodology for non-natural images and make the "latent augmentation" a common data augmentation technique in few-shot learning.


**Summary Of The Review:**

The paper shows very promising results for few-shot learning in histology images and introduces a simple and effective method for doing data augmentation. The paper would benefit from some additional experiments with competing augmentation methods and a few straightforward ablations. Overall, the paper is very well written, feels complete, and has an extensive discussion.

---

> ### Author Response · Authors · 2021-11-19
> **Response to Reviewer hJxe (1/2)**
>
> We sincerely thank Reviewer hJxe for the detailed and valuable comments and the recognition of our efforts.
>
> Below we address each concern separately.
>
> > Q1. Why didn't you compare with other " variation augmentation" methods? For example, how well does adding white noise to the latent variables perform?
>
> **A1:** Thanks for pointing out this valuable point, which is also suggested by the Reviewer ijae.
>
> Following the suggestion, we conduct an additional ablation study to explore more types of covariances. We use notations of GMM in scikit-learn library [1], and define more covariance types:
> 1) ``Tied``, where all clusters share a covariance matrix estimated from the entire base dataset.
> 2) ``Diag``, where each cluster has its own diagonal covariance matrix, i.e. diagonal elements are a variance vector and non-diagonal elements are zeros
> 3) ``Spherical``, where each cluster has its own single scalar-variance shared by all feature dimensions.
>
> Our default setting is ``Full``. The table below reports the results in the ablation task.
>
> | Cov Type      | Base       | Novel  | HMean|
> |:--------------:|:----------:|:----------:|:----------:|
> | ``Baseline``  | 85.85±0.78 | 53.27±1.63 | 65.74±1.06 |
> | ``Tied``          | 79.35±1.08 | *65.32±1.21* | 71.65±1.14 |
> | ``Diag``          | *85.91±0.88* | 62.66±1.42 | *72.46±1.08*|
> | ``Spherical``     | 85.78±0.87 | 62.00±1.39 | 71.97±1.07 |
> | ``Full (default)``| **87.37±0.81** | **66.51±1.39** | **75.11±1.01** |
>
> LA improves baseline with all types of covariance with respect to Harmonic mean metrics. Using the covariance estimated from the entire dataset, i.e. ``Tied``, however, hurts base class performance.  Our default setting (``Full``) achieves the best performance.
>
>
> > Q2. Add a small investigation of the shape of the clusters (i.e., the covariance matrix). K-Means corresponds to GMM with an isotropic covariance matrix. If it turns out that isotropic covariance matrices are sufficient, this information would be helpful in scenarios with small clusters (when the covariance approximations are poor).
>
> **A2:** Thanks for the insightful suggestion! We have not come to this point before. The table in **A1** reports the results of our default setting, where 16 clusters are used. Here we further explore more options of cluster numbers (leading to larger or smaller clusters). Figure D.2 in our revised paper shows the results. LA with full covariance matrix does degenerate performance as the cluster size decreases (larger number of prototypes). In contrast, LAs with isotropic covariance matrices (``Spherical``/``Diag``, if we understood correctly) perform more stably when cluster size alters. However, they underperform ``Full`` covariance in all cluster sizes. Besides, similar performance degeneration is also observed but becomes slight. Overall, the ``Full`` covariance matrix is preferable based on our current observation.
>
> > Q3. How stable is the method to the randomness in K-means? Did you notice any significant performance variation when running K-means with different random seeds?
>
> **A3:** we use a fixed random seed for reproducibility during our experiments. Now, we add more experiments during the rebuttal period to confirm LA's robustness.
>
> The table below reports the results of the same ablation task specified in our paper (300 meta-tasks under the 5-shot setting) using different random seeds. Their numerical differences are within a 95% confidence interval, i.e. LA performs stably under different random seeds. We have revised our paper accordingly to demonstrate this point in Section 4.3 **Additional ablation studies** and Appendix D.4.
>
>
> | Random Seed   | Base       | Novel  | HMean|
> |:--------------:|:----------:|:----------:|:----------:|
> | Baseline      | 85.85±0.78 | 53.27±1.63 | 65.74±1.06 |
> | 10            | 86.85±0.84 | 65.70±1.40 | 74.81±1.05 |
> | 20            | 87.34±0.80 | 66.74±1.33 | 75.66±1.00 |
> | 30            | 87.71±0.81 | 66.48±1.35 | 75.63±1.01 |
> | 40            | 87.31±0.80 | 65.04±1.39 | 74.55±1.02 |
> | 50            | 87.83±0.78 | 66.40±1.35 | 75.63±0.99 |
> | 66 (default)  | 87.37±0.81 | 66.51±1.39 | 75.53±1.03 |
> | Average over 6 seeds | 87.40±0.81 | 66.14±1.37 | 75.30±1.01|

---

> > ### Author Response · Authors · 2021-11-19
> > **Response to Reviewer hJxe (2/2)**
> >
> >
> > ### minor comments.
> >
> > > Q4. The experiment (and the caption) of figure 4 is unclear, even though the appendix provides many details. Please refactor the explanation so that it doesn't require reading the appendix (for example, explain what the features are inside the caption).
> >
> > **A4:** Sorry for the confusion, we now update the discussion of "Why do CLP models generalize better than FSP ones in histology?" in our revised paper. Figure 4 is mainly discussed there. We replace "feature" with "feature map" to specify its shape. A feature map extracted from a ResNet before average pooling should be $F^{H\times W \times C}$, e.g. given an input image of size 448x448, a ResNet-18 typically produces a 14x14x512 feature map from its last stage (high-level), and a 28x28x256 feature map from its penultimate stage (middle-level).
> >
> >
> > > Q5. It's unclear how l2 feature normalization is used in clustering and downstream meta-tasks. As of now, I understand that l2 feature normalization is applied before doing clustering. However, all feature vectors would be on the unit hypersphere, giving a meaningless cluster covariance matrix. Please clarify.
> >
> > **A5:** Yes, we do clustering and estimate cluster covariance matrix on $l_2$-normalized features. After all $l_2$-normalization, all original samples are on the surface of a unit-hypersphere (a $d-1$ dimensional manifold, say $\mathcal{S}^{d-1}$). To inspect whether the covariance is still meaningful after normalization, we do additional ablation in Appendix D.5 of the revised paper. Specifically, we compute the covariance matrices before and after $l_2$-normalization and find they are highly correlated, e.g. their Pearson correlation coefficients > 0.99 for almost all cases. Thus, the covariances may still be informative to some extend. Besides, a recent work on self-supervised learning that studies variance and invariance in non-contrastive learning, BarlowTwins [2], also computes covariance after $l_2$-normalization in their ablation study (Section 4, "Loss Function Ablations" in [2]).
> >
> > We also do a small additional ablation: since after $l_2$-normalization, all original samples are on the surface of a unit-hypersphere but the augmented samples generated by LA are not likely to be on that surface. We thus further normalize the augmented samples to enforce them to be on that surface. In the same ablation task, we observe the performance is marginally affected from 75.30%+-1.01% to 74.69%+-1.01% if we normalize augmented samples. The change is within a 95% confidence interval. Although the $l_2$-normalized features are on the surface of a unit-hypersphere, the linear classifiers, typically, are still fitted in the original $\mathbb{R}^d$ space and no special consideration is taken. Generating augmented samples that are not on the surface of the unit-hypersphere may still help to some extent. More discussion about the niceness of unit-hypersphere is also included in the discussion section in [3].
> >
> > > Q6. Writing and typos suggestions.
> >
> > **A6:** Thanks for the correction and suggestions! We have revised them accordingly. In particular, we change "legacy dictionary" to "base dictionary". "Space is linearized": the deep features are highly linearized so that linearly interpolating or extrapolating them are likely to give meaningful features. We have revised the text accordingly. "+- numbers in Table 1": they specify the confidence interval. The caption of Table 1 has been modified to notify this point.
> >
> > > Q7. Will the code be released?
> >
> > **A7:** Yes, we will release our code upon paper publication.
> >
> > Again, we sincerely appreciate the detailed comments and your recognition of our efforts!
> >
> > ----------
> > ### Reference
> >
> >
> > [1] sklearn.mixture.GaussianMixture https://scikit-learn.org/stable/modules/generated/sklearn.mixture.GaussianMixture.html
> >
> > [2] Zbontar, Jure, et al. "Barlow twins: Self-supervised learning via redundancy reduction." arXiv preprint arXiv:2103.03230 (2021). https://arxiv.org/abs/2103.03230
> >
> > [3] Wang, Tongzhou, and Phillip Isola. "Understanding contrastive representation learning through alignment and uniformity on the hypersphere." International Conference on Machine Learning. PMLR, 2020. https://arxiv.org/abs/2005.10242

---

> > > ### Comment · Reviewer_hJxe · 2021-11-21
> > > **More details addressing Q1**
> > >
> > > I thank the authors for addressing almost all my questions, and I welcome the additional ablations they carried and added to the paper.
> > >
> > > I would like more details wrt Q1: "Why didn't you compare with other " variation augmentation" methods?", because I didn't receive an answer. While I welcome the ablation studies with different covariance types, I am curious how other similar methods compare to LA.
> > >
> > > On page 9, before the conclusion, you list several "representation variation augmentation approaches". While I understand your approach is different (e.g., it doesn't require labels), I am curious how the LA method compares to these methods. For example, which existing methods would you call "direct competitors" to your LA method? How does LA compare to these methods?

---

> > > > ### Author Response · Authors · 2021-11-22
> > > > **Discussion with Reviewer hJxe on Q1**
> > > >
> > > > We thank you for your careful reviews and time on our work.
> > > >
> > > > We apologize for the misunderstanding about Q1. In the first round response, we thought we answer the question by answering *"For example, how well does adding white noise to the latent variables perform?"*. We see ablations with different covariance types, e.g. using the spherical covariance, as adding meaningful noise to the latent variables.
> > > >
> > > > Below, we address the remaining part of Q1 in detail.
> > > >
> > > > The most related representation variation augmentation approch to ours is Distribution Calibration (DC) [1], where it adopts meta-training as the pre-training step and uses label information to compute the base classes statistics (mean, covariance), i.e. the "base dictionary" in our paper. Then, DC calibrates novel classes' statistics. Specificially, they compute $\mu'=(\sum_{i\in \mathbb{S}} \mu_i + z)/(k+1)$, $\Sigma'=(\sum_{i\in \mathbb{S}} \Sigma_i)/k + \alpha$. The augmented samples are then generated from $\mathcal{N}(\mu',\Sigma')$. Here, $\mathbb{S}$ is the top-$k$ similar classes set, i.e. $|\mathbb{S}|=k$. $k$ and $\alpha$ are two hyper-parameters. In their experiments, they find calibrating from two base classes, i.e. $k=2$ achieve the best results.
> > > >
> > > > Despite the unfairness of using label information, here we show the results using DC in our experiments. We set $k=1, \alpha=0$ (the number of base classes in our problem is 6, significantly smaller than 64, 160, and 100 classes in natural images[1]; we thus use k=1).
> > > >
> > > > | Method   | Base       | Novel  | HMean|
> > > > |:--------------|:----------:|:----------:|:----------:|
> > > > | DC             | 90.33±0.61 | 70.01±1.24 | 78.88±0.82 |
> > > > | LA + supervised base dictionary | 90.51±0.59 | 68.37±1.31 | 77.90±0.81 |
> > > > | LA w/. 16-prototype (default) | 87.37±0.81 | 66.51±1.39 | 75.11±1.01|
> > > > | LA w/. 6-prototype            | 86.07±1.31 | 68.30±1.31 | 76.16±1.03 |
> > > > | LA w/. 6-prototype + calibration | 85.41±0.88 | 69.05±1.22 | 76.36±1.02|
> > > >
> > > > In the table above, DC = LA + supervised base dictionary + calibration.
> > > >
> > > > By explicitly exploiting label information, "DC" and "LA + supervised base dictionary" achieve the best results. Using the prior label distribution knowledge (i.e. knowing there are 6 classes in the base training dataset), "LA w/. 6-prototype base dictionary" can achieve better results than our default setting. Further calibration (i.e. sampling from $\mathcal{N}((\mu_{i^*} + z)/2, \Sigma_{i^\star})$, instead of $\mathcal{N}(\mu_{i^*}, \Sigma_{i^\star})$) marginally improves LA when using unsupervised constructed base dictionary  (76.16-->76.36). It improves more when using the supervised base dictionary (77.90-->78.88).
> > > >
> > > > Our default number of prototypes is obtained by ablation study, assuming no label information is exposed, including the number of classes in the base dataset.
> > > >
> > > > In related work, Hariharan & Girshick [2] adopt a generator to generate "hallucinated" novel features from the variation of given base samples; Wang et al. [3] generate "hallucinated" from a given sample and a randomly sampled noise vector. They require training an additional generator during the pre-training stage. We also notice some recent works that employ a similar idea of "learning to augment from the others", e.g. [4]. However, the aforementioned methods in this paragraph have not made their codes available yet. We will try to include the results if we successfully re-implement their methods or if we find reliable re-implementations.
> > > >
> > > > Other methods using the idea of representation variation are specified for other applications, e.g. Wang et al.[5,6] use LA to improve performance in standard classification and segmentation by deriving the upper bound when optimizing cross-entropy loss with the augmented feature $\hat{z}\sim \mathcal{N}(z, \lambda \Sigma_{y})$ ($\lambda$ is a positive coefficient to control the strength of augmentation); Yin et al. [7] and Liu et al. [8] use LA to transfer the covariance of "head" classes to "tail" classes in a similar way (Yin et al. use PCA extract the principle components in covariance matrix while Liu et al. use AMSoftmax and directly sample from multivariate Gaussian), so as to improve the under-representative classes in long-tailed recognition problems. Since they are studying different problems, we exclude them from the compared methods.

---

> > > > > ### Author Response · Authors · 2021-11-22
> > > > > **Reference for discussion**
> > > > >
> > > > >
> > > > > ----
> > > > > ### Reference
> > > > > [1] Yang, Shuo, Lu Liu, and Min Xu. "Free lunch for few-shot learning: Distribution calibration." arXiv preprint arXiv:2101.06395 (2021). [paper link](https://arxiv.org/pdf/2101.06395).
> > > > >
> > > > > [2] Hariharan, Bharath, and Ross Girshick. "Low-shot visual recognition by shrinking and hallucinating features." Proceedings of the IEEE International Conference on Computer Vision. 2017. [paper link](https://openaccess.thecvf.com/content_ICCV_2017/papers/Hariharan_Low-Shot_Visual_Recognition_ICCV_2017_paper.pdf).
> > > > >
> > > > > [3] Wang, Yu-Xiong, et al. "Low-shot learning from imaginary data." Proceedings of the IEEE conference on computer vision and pattern recognition. 2018. [paper link](https://openaccess.thecvf.com/content_cvpr_2018/papers/Wang_Low-Shot_Learning_From_CVPR_2018_paper.pdf).
> > > > > [4] Park, Seong-Jin, et al. "Meta variance transfer: Learning to augment from the others." International Conference on Machine Learning. PMLR, 2020. [paper link](http://proceedings.mlr.press/v119/park20b/park20b.pdf).
> > > > >
> > > > > [5] Wang, Yulin, et al. "Implicit semantic data augmentation for deep networks." Advances in Neural Information Processing Systems 32 (2019): 12635-12644. [paper link](https://proceedings.neurips.cc/paper/2019/file/15f99f2165aa8c86c9dface16fefd281-Paper.pdf).
> > > > >
> > > > > [6] Wang, Yulin, et al. "Regularizing deep networks with semantic data augmentation." IEEE Transactions on Pattern Analysis and Machine Intelligence (2021). [paper link](https://ieeexplore.ieee.org/stamp/stamp.jsp?arnumber=9332260&casa_token=W3ycYRnPEogAAAAA:0sGyynX4NGUjPqB9w9nL24-L5WjfSZivGQZMs8Ym0CDp3R_iNdvNJEqUIkeXyX-1pZcZt1RtKw&tag=1).
> > > > >
> > > > > [7] Yin, Xi, et al. "Feature transfer learning for face recognition with under-represented data." Proceedings of the IEEE/CVF Conference on Computer Vision and Pattern Recognition. 2019. [paper link](https://openaccess.thecvf.com/content_CVPR_2019/papers/Yin_Feature_Transfer_Learning_for_Face_Recognition_With_Under-Represented_Data_CVPR_2019_paper.pdf).
> > > > >
> > > > > [8] Liu, Jialun, et al. "Deep representation learning on long-tailed data: A learnable embedding augmentation perspective." Proceedings of the IEEE/CVF Conference on Computer Vision and Pattern Recognition. 2020. [paper link](https://openaccess.thecvf.com/content_CVPR_2020/papers/Liu_Deep_Representation_Learning_on_Long-Tailed_Data_A_Learnable_Embedding_Augmentation_CVPR_2020_paper.pdf).

---

> > > > > > ### Comment · Reviewer_hJxe · 2021-11-22
> > > > > > **Answer to comparison with "Distribution Calibration"**
> > > > > >
> > > > > > I thank the authors for getting back with new experiments in less than 24h. I assume that it's expected that DC provides slightly better results than LA, because it uses label information.
> > > > > >
> > > > > > Can we conclude that one should use DC when accessing labeled data and use LA when there is significant unlabelled data? Please add the results comparing LA with DC to the paper, and clarify whether other methods (e.g., DC) should be used when accessing labeled information.
> > > > > >
> > > > > > After you make these changes I concluded all my concerns with the paper, and I will keep my score to 8 (accept) because the paper addresses an interesting topic (histology images), showing that constructive learning gives significant improvement ~+10%, and the proposed latent augmentation can provide an additional ~10% performance.

---

> > > > > > > ### Author Response · Authors · 2021-11-23
> > > > > > > **Concluding with Reviewer hJxe**
> > > > > > >
> > > > > > > We have revised our paper to include the discussion comparisons to distribution calibration in the ablation study as well as in Appendix.
> > > > > > >
> > > > > > > We sincerely appreciate your valuable comments, your time and your efforts in reviewing and improving our work!

---

### Official Review · Reviewer_PqMe · 2021-11-02

**Correctness:** 4
**Technical Novelty And Significance:** 3
**Empirical Novelty And Significance:** 2
**Recommendation:** 5
**Confidence:** 3

**Main Review:**

Strength:
1. The proposed LA is interesting and shows superior data augmentation including RandomResizedCrop, RandomHorizontalFlip, and ColorJitter.
2. The paper is rich in contents and provided a comprehensive appendix.
3. The study of few shot learning on histological images are at its early stage. This paper's result show that CL can be better than supervised pretraining, which is a encouraging finding. It will be inspiring for researchers in this field.

Weakness:
1. The technical novelty of this paper is limited. The CL strategy is the existing MoCo v3. LA is simple and effective, but it has supervised counterparts (as stated in the related work part).
2. Lack of comparisons. The authors compared "CL vs. supervised learning" and "LA vs. stardard data augmentation", but did not compare with other few-shot learning, semi-supervised learning and data augmentation algorithms. These methods can also be used in problems with few labeled samples.
3. In Table 1, LA is compared with no data augmentation (if I understand correctly), which is unfair because data augmentation have become a standard step, so LA should be compared with other data augmentation methods.
4. Other self-supervised learning (SSL) papers, such as "Unsupervised Learning of Dense Visual Representations, NeurIPS 2020", "Dense contrastive learning for self-supervised visual pre-training, CVPR 2021", "Momentum Contrast for Unsupervised Visual Representation Learning, CVPR 2020", have found that SSL sometimes outperform supervised pretraining. Therefore, the finding of this paper is not so surprising. In this paper, the novel classes are improved more by SSL pretraining, while supervised pretraining still prevails in the base classes, which is consistent with existing papers which found that SSL are better when the task or dataset is different in pretraining and finetuning.
5. The paper is rich in contents and provided a comprehensive appendix. However, it may make the main body not very self-contained. It may be better to submit the full paper to a journal in this field such as Medical Image Analysis. For example, the core algorithm, MoCo-v3, should be explained in the main body.

Question: Is self-supervised learning performed on manual crops of WSIs? Since each patch may contain one manually assigned label, does that mean labels have been somehow encorporated in the self-supervised learning process? Ideally, self-supervised learning should be performed on WSIs without manual cropping.

**Summary Of The Paper:**

The paper proposes to use contrastive learning (CL) with a novel data augmentation (latent augmentation, LA) strategy to build a few-shot system for histology image classification. Two empirical findings are: i) models learned by CL generalize fairly better than supervised learning for histology images, and ii) LA brings consistent gains over baselines without data augmentation. Analyses are given to discuss these findings.

**Summary Of The Review:**

The paper has slight novelty and interesting findings but neither novelty nor finding is significant enough. It may be better to be submit it to medical image journals.

---

> ### Author Response · Authors · 2021-11-19
> **Response to Reviewer PqMe (1/2)**
>
> We thank Reviewer PqMe for the comments and the positive feedback on our efforts.
>
> Below we address each concern separately.
>
> > Q1. The technical novelty of this paper is limited. The CL strategy is the existing MoCo v3. LA is simple and effective, but it has supervised counterparts (as stated in the related work part).
>
> **A1:** One of our major technical contributions is to show the potential of the use case of latent augmentation in a totally unsupervised manner or in a self-supervised manner to some extent. To our best knowledge, this unsupervised use case may have not been verified or widely studied before, as also pointed out by other reviewers. We are not trying to overclaim our contributions but tried our best to give sufficient credit to previous works as mentioned by the reviewer, e.g. MoCo-v3 and supervised variation methods that are clearly stated in our related work section.
>
> Another vital and intriguing contribution of this work is to show how few-shot classification in histology images can benefit from the combination of contrastive learning pre-training (CLP) and a self-supervised latent augmentation (LA) method, with decent gains.
>
> We notice the large generalization gap between fully-supervised pre-training (FSP) and CLP in our problem. Thereafter, we try to analyze what might contribute to better generalizability. That's why we entitle our work part as "***Towards better understanding and better generalization***". Since few-shot learning in histology images is at its early stage, and self-supervised learning is a fast-growing field, we hope this work would provide useful data points and experience for them, from a representation learning perspective (more discussion about empirical findings and why they matter in **A4**.)
>
> > Q2&3. (Q2) Lack of comparisons. The authors compared "CL vs. supervised learning" and "LA vs. standard data augmentation", but did not compare with other few-shot learning, semi-supervised learning, and data augmentation algorithms. These methods can also be used in problems with few labeled samples. (Q3) In Table 1, LA is compared with no data augmentation. Data augmentation has become a standard step, so LA should be compared with other data augmentation methods.
>
> **A2&3:**
> * Comparison to other works: as stated in the paper (Sec. 4.1 ***Compared methods.***), recent works [1][2][3] find that various sophisticated episodic training (meta-training) few-shot learning methods are no better than simple pre-training & fine-tuning. Hence, we also follow this scheme and make comparisons under this setting.
>
> * LA v.s. DA: we compare data augmentation (DA) and LA in the ablation study (Fig. 3(b) and Fig. D.1 in appendix). Results show that DA's gains are not consistent (degenerated base class performance, Fig. D.1) and saturate earlier. In contrast, LA's improvement persists in both base classes and novel classes. Table 1 reports results averaged from 9 sub-tasks (near-domain task). To get a sweet spot from data augmentation, according to Figure-3(b), a 49 ~ 64 times more computation budget (image augmentation in RGB space and encoder feedforwards) is needed. This brings significantly more computation budget but, likely, no better results than LA. Thus, we compare DA and LA in the ablation study only.

---

> > ### Author Response · Authors · 2021-11-19
> > **Response to Reviewer PqMe (2/2)**
> >
> > > Q4. Other self-supervised learning (SSL) papers have found that SSL sometimes outperforms supervised pretraining. Therefore, the finding of this paper is not so surprising and is consistent with existing papers.
> >
> > **A4:** Indeed, SSL methods outperform supervised pre-training in many fields. But there are more topics worth exploring for SSL, e.g. studying what makes good views for contrastive learning [4], what should not be contrastive in contrastive learning [5], invariance and equivariance in contrastive learning [6], and what makes instance discrimination good for transfer learning [7], to list a few.
> >
> > Previous SSL works mainly focus on pre-training in natural images, especially in the ImageNet dataset, a large-scale dataset with diverse classes and diverse samples. We see our work as a useful data point for studying FSP and CLP in one type of non-standard natural image, histology images. As also noted by the Reviewer hJxe, we would like to re-emphasize the **large** generalization gap (~+10% F1 score) between CLP and FSP shown in **our problem** compared to those in standard natural images. We then investigate why the large gap exists through visualization, and provide our empirical understandings in Section 4.4 (**Why do CLP models generalize better than FSP ones in histology images?**) in our revised paper. We should point out that our discussion there is about what the reasons behind this large gap can be. The large gap in our problem still remains as an empirical observation. We believe the observation leave room for future research on studying FSP and CLP, for example:
> > * Is it the label diversity of pre-training datasets that causes a larger generalization gap? FSP and CLP models are both pre-trained on a  9-class dataset (NCT dataset) in our problem. Previous works train FSP and CLP models on a 1000-class dataset (ImgNet dataset). The label diversity of ImgNet is significantly larger than NCT. Will this impact the FSP models' generalizability as we hypothesize at the end of the discussion section (in the revised paper)?
> > * Is the sample diversity that causes a larger generalization gap? histology patches usually have a similar appearance and have less diversity compared to natural images.
> > * Is it the dataset scale that causes a larger generalization gap? NCT dataset has 100k samples while ImgNet has ~ 1.28M samples.
> > * Is it the histology image itself that causes a larger generalization gap? histology images usually have repeated patterns and textures and lack dominant objects, which makes them different from natural images.
> >
> > Thus, we believe findings and discussions in our paper could contribute to certain topics in representation learning and could be appealing to the audiences of the ICLR conference.
> >
> >
> > > Q5. The paper is rich in contents and provided a comprehensive appendix. However, it may make the main body not very self-contained. It may be better to submit the full paper to a journal in this field such as Medical Image Analysis. For example, the core algorithm, MoCo-v3, should be explained in the main body.
> >
> > **A5:** Given the answers to Q4, we think our work is worth a spot at the ICLR conference. MoCo-v3 is a running example of contrastive methods and its details are not the major interest of our work. Therefore we defer it to the appendix.
> >
> >
> > > Q6: Is self-supervised learning performed on manual crops of WSIs? Since each patch may contain one manually assigned label, does that mean labels have been somehow encorporated in the self-supervised learning process? Ideally, self-supervised learning should be performed on WSIs without manual cropping.
> >
> > **A6:** Thanks for raising a valuable topic. Currently, we pre-train our model on the NCT dataset, which is "clean" and fine-grained to some extent. Exploring self-supervised learning in real-world, uncurated, long-tailed datasets like real WSIs is no doubt the next step. Similar problems also occur in ImageNet pre-training. All images in ImageNet contain one manually assigned label that might have been somehow incorporated in the self-supervised learning process. Thus, some recent works also explore self-supervised pre-training in the uncurated wild dataset, e.g. [8].

---

> > > ### Author Response · Authors · 2021-11-19
> > > **Missing Reference**
> > >
> > > ### Reference
> > > [1] Chen, Wei-Yu, et al. "A Closer Look at Few-shot Classification." International Conference on Learning Representations. 2018. https://arxiv.org/abs/1904.04232
> > >
> > > [2] Tian, Yonglong, et al. "Rethinking few-shot image classification: a good embedding is all you need?." Computer Vision–ECCV 2020: 16th European Conference, Glasgow, UK, August 23–28, 2020, Proceedings, Part XIV 16. Springer International Publishing, 2020. https://arxiv.org/pdf/2003.11539
> > >
> > > [3] Shakeri, Fereshteh, et al. "FHIST: A Benchmark for Few-shot Classification of Histological Images." (2021).
> > >
> > > [4] Tian, Yonglong, et al. "What makes for good views for contrastive learning?." arXiv preprint arXiv:2005.10243 (2020). https://arxiv.org/abs/2005.10243
> > >
> > > [5] Xiao, Tete, et al. "What should not be contrastive in contrastive learning." arXiv preprint arXiv:2008.05659 (2020). https://arxiv.org/pdf/2008.05659.pdf
> > >
> > > [6] Dangovski, Rumen, et al. "Equivariant Contrastive Learning." arXiv preprint arXiv:2111.00899 (2021). https://arxiv.org/abs/2111.00899
> > >
> > > [7] Zhao, Nanxuan, et al. "What makes instance discrimination good for transfer learning?." arXiv preprint arXiv:2006.06606 (2020). https://arxiv.org/pdf/2006.06606.pdf
> > >
> > > [8] Goyal, P., Caron, M., Lefaudeux, B., Xu, M., Wang, P., Pai, V., ... & Bojanowski, P. (2021). Self-supervised pretraining of visual features in the wild. arXiv preprint arXiv:2103.01988. https://arxiv.org/abs/2103.01988

---

> > > > ### Comment · Reviewer_PqMe · 2021-11-23
> > > > **The paper is more suitable for a medical imaging venue**
> > > >
> > > > I appreciate the authors' elaboration on the contribution of the paper and the insightful discussion on the question "Why do CLP models generalize better than FSP ones in histology images". I also noticed the significant gain brought by the proposed LA, as also pointed out by other reviewers.
> > > >
> > > > However, if the main contribution of this paper is the novel data augmentation method LA, there should be more comparisons with other SOTA few-shot learning, semi-supervised learning and data augmentation algorithms, instead of simply comparing with standard data augmentation. If the main contribution is the novel algorithm, experiments on other datasets should also be conducted. A good example is the paper "Free lunch for few-shot learning: Distribution calibration" accepted by ICLR 2021.
> > > >
> > > > On the other hand, if the main contribution is the findings limited on the field of histology images, the paper will be more suitable for a medical imaging venue. Otherwise, the authors must demonstrate the finding applies to more than one datasets, so as to meet the high standard of ICLR.

---

> > > > > ### Author Response · Authors · 2021-11-24
> > > > > **Response to and discussion with Reviewer PqMe (1/3)**
> > > > >
> > > > > We thank reviewer PqMe for the responses, especially the recognition of our discussion on the question: "Why do CLP models generalize better than FSP ones in histology images", which we also believe could be insightful and intriguing for audiences in the ICLR community.
> > > > >
> > > > >
> > > > > However, reviewer PqMe has raised two major concerns. The followings are our responses.
> > > > >
> > > > > > Q1: There should be more comparisons with other SOTA few-shot learning, semi-supervised learning, and data augmentation algorithms, instead of simply comparing with standard data augmentation.
> > > > >
> > > > > **A1:**
> > > > >
> > > > > This concern has been raised in the initial review comments. We tried to answer it under A2&3 in our previous responses. We are sorry that our responses are not convincing to the reviewer. We would like to elaborate more below.
> > > > >
> > > > > **Comparison with other representation variation augmentation methods.**
> > > > >
> > > > > As noted by and discussed with Reviewer hJxe, our approach is different from prior arts in "doesn't require labels". It does not require labels in both the pre-training step and the latent augmentation (LA) step. This makes it hard for an eventually fair comparison, as other related methods usually require labels.
> > > > >
> > > > > Despite the unfairness of using label information, in our latest revision, we have included the results from the supervised distribution calibration (DC) ("Free lunch for few-shot learning: Distribution calibration" accepted by ICLR 2021. [12]). It turns out that the unsupervised LA can perform comparably well with the supervised DC. We see it as a positive sign for studies in unsupervised and self-supervised learning. The label-hungry issue exists almost everywhere, which is one of the reasons why self-supervised learning is becoming more popular in recent years, and why the passion for unsupervised learning never fades. Also, please feel free to refer to "Discussion with Reviewer hJxe on Q1" for an elaboration on the topic "compare with other representation variation augmentation methods".
> > > > >
> > > > >
> > > > > **Comparison with other few-shot learning methods.**
> > > > >
> > > > > We see standard pre-training & fine-tuning as a strong baseline that can achieve comparable or even better performance than other recent few-shot learning methods. Recent works [1,2] and concurrent works [3,4] find the simple "whole-classification" pre-training (i.e. not splitting the dataset to episodic meta-tasks) and fine-tuning can provide better results than many recently proposed few-shot meta-learning algorithms. This observation is further studied and confirmed by an ICCV 2021 work [5]. We thus followed this trend to conduct our experiments.
> > > > >
> > > > > Besides, we later found that the advantage of the standard "whole-classification" pre-training is also observed in a simultaneously concurrent benchmark work on histology few-shot classification study [6]. In that work, they compare ProtoNet[7], MetaOpt[8], SimpleShot[2], Distill[4], Finetune[3], MAML[9], LaplacianShot[10], and TIM[11]. Only the transductive TIM [11] can have better results than the inductive simple pre-training and fine-tuning (Finetune[3], Distill[4]). With such empirical evidence in both natural images and histology images, we thus use pre-training + different fine-tuned classifiers to build our comparisons. Our experiments reproduce the interesting observations in prior works, e.g. NearestNeighbor classifier sometimes yields better results than linear classifiers. Prior arts [2,4] have observed it and tried to analyze it from different perspectives.
> > > > >
> > > > > **Comparison with semi-supervised learning methods and other data augmentation methods.**
> > > > >
> > > > > Thanks for raising this topic! To our knowledge, extending existing semi-supervised methods (e.g. mixmatch, fixmatch, remixmatch) and other data augmentation methods (e.g. mix-up, UDA) into few-shot classification is a non-trival work. We have not noticed related works yet. It would be great if Review PqMe can help us in listing some open-source related works with direct applications in few-shot classification. We could try to include them in the camera-ready version.

---

> > > > > > ### Author Response · Authors · 2021-11-24
> > > > > > **Response to and discussion with Reviewer PqMe (2/3)**
> > > > > >
> > > > > > > Q2.1: The authors must demonstrate the finding applies to more than one datasets.
> > > > > >
> > > > > > Our experiments are conducted on 3 independent datasets from different sources. They vary in terms of organs, stainings, and imaging protocols. We aim to study the generalization problem, which is why we define three tasks that simulate different degrees of domain shift. Thus, we do demonstrate the finding applies to more than one dataset. If the question means applying to more than one type of dataset, we have more discussion later at the bottom.
> > > > > >
> > > > > > > Q2.2: The paper will be more suitable for a medical imaging venue.
> > > > > >
> > > > > > When looking at the [About Us](https://iclr.cc/) in the ICLR website and another [About](https://iclr.cc/About) ICLR page, we can find the following descriptions:
> > > > > >
> > > > > > > ICLR is globally renowned for presenting and publishing cutting-edge research on all aspects of deep learning used in the fields of [...] as well as ***important application areas*** such as machine vision, ***computational biology*** [.....].
> > > > > > >
> > > > > > > A non-exhaustive list of relevant topics explored at the conference include:
> > > > > > > ...
> > > > > > >
> > > > > > > applications in audio, speech, robotics, ***neuroscience, computational biology***, or any other field
> > > > > > > ...
> > > > > >
> > > > > > As ICLR is open to topics like neuroscience and computational biology, we believe applications in the medical domain should not be excluded. And this can be verified by exploring the historical articles in ICLR, e.g. :
> > > > > >
> > > > > > * Vig, Jesse, et al. "BERTology Meets Biology: Interpreting Attention in Protein Language Models." ICLR 2021.
> > > > > > * Qi, Gege, et al. "Stabilized Medical Image Attacks." ICLR 2021.
> > > > > > * Jarrett, Daniel, et al. "Clairvoyance: A pipeline toolkit for medical time series." ICLR 2021.
> > > > > > * Angermueller, Christof, et al. "Model-based reinforcement learning for biological sequence design." ICLR 2020.
> > > > > >
> > > > > > to list a few. Moreover, our focus in this study is the few-shot learning for histology images, which ultimately lands in learning representations of histology images. And as the reviewer PqMe agrees that our discussion in "Why do CLP models generalize better than FSP ones in histology images" is insightful, we do believe this work can be intriguing for interested audiences in not only histology image analysis community, but also self-supervised learning community, representation learning community, visual domain generalization community, and beyond. We thus believe this paper should be of major interest to ICLR audiences, as agreed by other reviewers.
> > > > > >
> > > > > >
> > > > > > Also, we would like to share further thoughts on this issue beyond the scope of our paper:
> > > > > >
> > > > > > **Should learning the representation of medical images be presented on medical-related venues only?**
> > > > > >
> > > > > > We hope we can agree that learning medical image representation is equally important as other types of images, especially given the fact of the growing needs of healthcare and the shortage of medical staff. Machine learning can play a critical role in assisting pathologists and radiologists to ease the burden of reading images as well as improving the diagnosis accuracy. Given its importance, we believe the effort should not be limited to medical image analysis society. Thankfully, medical image applications have attracted sufficient attention from the machine learning community in recent years. In addition to ICLR, for top machine learning venues like NeurIPS, ICML, and top computer vision venues like CVPR, ECCV, ICCV, a considerable amount of studies on the medical/biological domain have been published every year and attracts audience's attention.

---

> > > > > > > ### Author Response · Authors · 2021-11-24
> > > > > > > **Response to and discussion with Reviewer PqMe (3/3)**
> > > > > > >
> > > > > > > **Is it a "must" for an ICLR paper to show a method applicable for different types of data?**
> > > > > > >
> > > > > > > We agree with the reviewer that it will be good to show the method is effective for different types of data. We also agree that having a method working on pathology images does not imply that it is applicable for other types of data. However, this problem also exists vise-versa, i.e. an algorithm effective for parts of nature images may not be effective for medical images. It does not mean that the studies based on ImageNet should be tested on medical images or satellite images to meet the standard of ICLR. Every method has its own limitation, and we believe as long as the scope of every proposed method is clearly stated, instead of being overstated without experimental supports, they should be fine.
> > > > > > >
> > > > > > > On the other hand, studies in different domains are inspiring each other in the past. For instance, U-Net is initially proposed and tested on medical image segmentation. But now it is widely adopted in the segmentation of all kinds of image data. Transformer is initially proposed for natural language processing. But now it is also applied in the computer vision domain, including pathology images. With a comprehensive analysis in this paper, as pointed out by Reviewer BmyB after our rebuttal, the topic we study could be of significant interest to the biomedical imaging community. Hopefully and ideally, our method should be applicable for other types of data, but that does need further verification in the future. We do plan to move forward to see whether this method generalizes to nature images, but the observations in this work leave room for further research, e.g. is the label diversity that causes the large generalization gap between FSP and CLP? is the label diversity that causes the large gain brought by unsupervised LA? Thus by introducing our work to the ICLR audience, we believe it can inspire not only medical image analysis but also much wider applications as recommended by other reviewers.
> > > > > > >
> > > > > > >
> > > > > > > Hope our responses can address the most of concerns raised by Reviewer PqMe. We appreciate the comments and do learn much.
> > > > > > >
> > > > > > > --------
> > > > > > >
> > > > > > > ### Reference
> > > > > > > [1] Gidaris, Spyros, and Nikos Komodakis. "Dynamic few-shot visual learning without forgetting." CVPR 2018.
> > > > > > >
> > > > > > > [2] Wang, Yan, et al. "Simpleshot: Revisiting nearest-neighbor classification for few-shot learning." arXiv preprint arXiv:1911.04623 (2019).
> > > > > > >
> > > > > > > [3] Chen, Wei-Yu, et al. "A Closer Look at Few-shot Classification." ICLR 2019
> > > > > > >
> > > > > > > [4] Tian, Yonglong, et al. "Rethinking few-shot image classification: a good embedding is all you need?." ECCV 2020.
> > > > > > >
> > > > > > > [5] Chen, Yinbo, et al. "Meta-Baseline: Exploring Simple Meta-Learning for Few-Shot Learning." ICCV 2021.
> > > > > > >
> > > > > > > [6] Shakeri, Fereshteh, et al. "FHIST: A Benchmark for Few-shot Classification of Histological Images." (2021).
> > > > > > >
> > > > > > > [7] Snell, Jake, Kevin Swersky, and Richard S. Zemel. "Prototypical networks for few-shot learning." arXiv preprint arXiv:1703.05175 (2017).
> > > > > > >
> > > > > > > [8] Lee, Kwonjoon, et al. "Meta-learning with differentiable convex optimization." CVPR 2019.
> > > > > > >
> > > > > > > [9] Finn, Chelsea, Pieter Abbeel, and Sergey Levine. "Model-agnostic meta-learning for fast adaptation of deep networks." ICML 2017.
> > > > > > >
> > > > > > > [10] Ziko, Imtiaz, et al. "Laplacian regularized few-shot learning." ICML 2020.
> > > > > > >
> > > > > > > [11] Boudiaf, Malik, et al. "Information Maximization for Few-Shot Learning." NeurIPS 2020.
> > > > > > >
> > > > > > > [12] Yang, Shuo, Lu Liu, and Min Xu. "Free lunch for few-shot learning: Distribution calibration." ICLR 2021.

---

> > > > > > > > ### Comment · Reviewer_PqMe · 2021-11-25
> > > > > > > > **Response to the authors**
> > > > > > > >
> > > > > > > > Thank you for your further clarification. There are two widely-accepted criteria for a paper to be accepted by a top CV/ML conference:
> > > > > > > >
> > > > > > > > 1. Novel and inspiring ideas. The idea may not be sophisticated, but it should inspire readers like "wow, I haven't thought it that way before!" However, in this paper, it is not so surprising to use unsupervised clusters to replace the label information in DC.
> > > > > > > >
> > > > > > > > 2. Significant accuracy improvement. To prove the improvement is significant, the method should be compared with a series of SOTA methods. However, the proposed method are only compared with DC and standard data augmentation including RandomResizedCrop, RandomHorizontalFlip, and ColorJitter. Among them, standard data augmentation is more of a baseline than a SOTA; LA is not better than DC. This means LA is not shown to be better than any SOTA.
> > > > > > > >
> > > > > > > > Actually I think LA is a good idea and it is much better than the baseline. I personally like the idea, but its efficacy needs more validation, which will make this paper a stronger paper. For example, the method DC was compared with 4 optimization-based, 5 metric-based, and 4 generation-based prior works. Perhaps it is harder to compare LA with so many works because the histology datasets are not so popular. However, it is by no means convincing to only compare with and outperform a baseline.
> > > > > > > >
> > > > > > > > Besides few-shot learning methods, there are also various data augmentation methods, such as "Generalizing Deep Learning for Medical Image Segmentation to Unseen Domains via Deep Stacked Transformation. TMI 2020", mix-up, "GAN-based synthetic medical image augmentation for increased CNN performance in liver lesion classification. Neurocomputing, 2018", etc. Can the authors explain why then cannot be compared in the settings of this paper? We can train the classifier on both the original representations z and the representations generated from augmented images.
> > > > > > > >
> > > > > > > > These are my two major critiques for the paper. I have been suggesting that this paper is more suitable for medical image venues, because I think their readers will be more interested in this paper. Many papers that published on MICCAI, TMI, MIA received more attention than those published on general CV/ML conferences, such as U-Net. Besides, on a medical journal the reviewers  (e.g. histology experts) will give more professional opinions, while the authors won't need to compress the main body and put so much useful contents in the appendix.

---

> > > > > > > > > ### Author Response · Authors · 2021-11-26
> > > > > > > > > **Addressing reviewer PqMe's concerns (1/2)**
> > > > > > > > >
> > > > > > > > > We thank the reviewer for the quick responses and the positive feedback on the idea of LA.
> > > > > > > > >
> > > > > > > > > We understand there is a disagreement between us and the reviewer on the contribution of unsupervised LA, but we would like to emphasize that our contributions are more than that, e.g.:
> > > > > > > > > * We facilitate the study of few-shot learning (FSL) in histology images. Also pointed out by the reviewer, FSL in histology is not so popular. That's one of our goals: to facilitate such study in histology image analysis and to provide an available benchmark for study in this domain. Learning representations of non-natural images also benefits the ICLR audiences, as they do show intriguing properties.
> > > > > > > > > * We show the CLP can generalize better than the FSP by a large margin in our problem. With further analysis, we provide our empirical understandings of this observation, which could be intriguing for studies in representation learning. Especially, we provide a useful data point for studies in why does CLP generalize better, and when it does. Since self-supervised learning and contrastive learning are still fast-growing areas, we believe such data points would be of great interest to not only the medical society but also the ICLR community.
> > > > > > > > >
> > > > > > > > > Below, we further address some specific concerns.
> > > > > > > > >
> > > > > > > > > >It is not so surprising to use unsupervised clusters to replace the label information in DC.
> > > > > > > > >
> > > > > > > > >
> > > > > > > > > We believe it is appealing to see the effectiveness of LA without labels. [DeepCluster](https://arxiv.org/abs/1807.05520), a self-supervised learning method, uses clustering to provide pseudo-labels as the classification targets for the next epochs. They replace the label information in ImageNet with unsupervised clustering, which is "not so surprising". But that does not hinder its contribution: showing how effective the unsupervised clustering can be. Showing the effectiveness of our approach is a positive sign: a border range of label-hungry applications could be encouraged to try CLP+LA for their problems. We understand you hold a different view on this contribution. Thus there is little we can rebuttal on this point and we would like to stop here and leave it open.
> > > > > > > > >
> > > > > > > > >
> > > > > > > > > > However, the proposed method are only compared with DC and standard data augmentation.
> > > > > > > > >
> > > > > > > > > We have elaborated our responses on the comparison problems in the **A1** of the second-round response.
> > > > > > > > >
> > > > > > > > > > LA is not better than DC. This means LA is not shown to be better than any SOTA.
> > > > > > > > >
> > > > > > > > > DC is not the lower bound of SOTAs, but a potential supervised upper bound: it has beaten 4 optimization-based, 5 metric-based, and 4 generation-based prior works. Showing LA is on par with DC could demonstrate LA is as good as other SOTAs, or even better as it requires no supervision, which is required by other SOTAs.
> > > > > > > > >
> > > > > > > > >
> > > > > > > > >
> > > > > > > > >
> > > > > > > > > > Comparison with baseline: "Is is **by no means** convincing to only compare with and outperform a baseline?"
> > > > > > > > >
> > > > > > > > > Sometimes the baseline is already competitive. That is what we argue in both two rounds of responses. Showing a method is better than a competitive baseline could be sufficient to demonstrate its effectiveness. For example, [DC](https://openreview.net/pdf?id=JWOiYxMG92s) uses [Mangla et al., (2020)](https://arxiv.org/pdf/1907.12087.pdf) as their baseline. If we look closer to this baseline (see the first row of Table 4 in [DC](https://openreview.net/pdf?id=JWOiYxMG92s)), the 5-way-5-shot performance there has already beaten 11 out of 13 state-of-the-art methods in Table 2. In our view, even if DC only compares with its baseline ([Mangla et al., (2020)](https://arxiv.org/pdf/1907.12087.pdf)) or uses weaker baselines, DC's merits do not get faded.  Yes, we agree that "it is harder to compare LA with so many works because the histology datasets are not so popular" since FSL in histology image is at the very early stage.

---

> > > > > > > > > > ### Author Response · Authors · 2021-11-26
> > > > > > > > > > **Addressing reviewer PqMe's concerns (2/2)**
> > > > > > > > > >
> > > > > > > > > > > Can the authors explain why the listed two papers cannot be compared in the settings of this paper?
> > > > > > > > > >
> > > > > > > > > > We thank the reviewer for this detailed suggestion. After reading the papers, we think they are not directly relevant to the few-shot classification problem studied in our work and it is less likely the methods in these two works will be as competitive as ours in solving the FSL problem:
> > > > > > > > > >
> > > > > > > > > > 1. ["Generalizing Deep Learning for Medical Image Segmentation to Unseen Domains via Deep Stacked Transformation. TMI 2020"](https://www.ncbi.nlm.nih.gov/pmc/articles/PMC7393676/):
> > > > > > > > > > This paper studies how to make a segmentation model trained on data from one source to be robust to data from other sources with simple but effective data augmentations. That said, this paper studies a domain adaption problem. However, in the domain adaption problem, the label space of the source domain and the target domain is usually the same, i.e. there are no **novel classes** in the target domain. In contrast, few-shot classification studies classifying novel classes that have not been exposed in the training stage. Domain adaption problem is usually formulated as an image-to-image translation problem. However, it is less likely if one can translate a pathology tissue type (base class) into the appearance of another one (novel class), i.e. the label space cannot be translated. The staining difference might be reduced with image-to-image translation. But color jittering can simulate the staining difference to some extent.
> > > > > > > > > >
> > > > > > > > > > 2. ["GAN-based synthetic medical image augmentation for increased CNN performance in liver lesion classification. Neurocomputing, 2018"](https://arxiv.org/pdf/1803.01229.pdf):
> > > > > > > > > > They train many DCGANs to synthesize liver lesion ROIs **for each lesion category separately** (on page 5 of their paper, Training Procedure), or train an ACGAN by explicitly passing the label information. During their training phase, samples from all classes are required. However, in few-shot learning, only the base classes are available for pre-training. No GAN can be trained to generate novel samples based on their paradigm.
> > > > > > > > > > Alternative relevant generator-based few-shot learning methods have been discussed in our related work section. They typically require specifically designed meta-training pipelines. For example, Hariharan & Girshick [1] adopt a generator to generate "hallucinated" novel features from the variation of given base samples; Wang et al. [2] generate "hallucinated" from a given sample and a randomly sampled noise vector. They require training an additional generator during the meta-training stage. We also notice some recent works that employ a similar idea of "learning to augment from the others", e.g. [3]. These methods are more few-shot-oriented, or meta-learning-oriented. However, they have not made their codes available yet. We will try to include the results if we successfully re-implement their methods or if we find reliable re-implementations.
> > > > > > > > > >
> > > > > > > > > >
> > > > > > > > > > [1] Hariharan, Bharath, and Ross Girshick. "Low-shot visual recognition by shrinking and hallucinating features." Proceedings of the IEEE International Conference on Computer Vision. 2017. [paper link](https://openaccess.thecvf.com/content_ICCV_2017/papers/Hariharan_Low-Shot_Visual_Recognition_ICCV_2017_paper.pdf).
> > > > > > > > > >
> > > > > > > > > > [2] Wang, Yu-Xiong, et al. "Low-shot learning from imaginary data." Proceedings of the IEEE conference on computer vision and pattern recognition. 2018. [paper link](https://openaccess.thecvf.com/content_cvpr_2018/papers/Wang_Low-Shot_Learning_From_CVPR_2018_paper.pdf).
> > > > > > > > > >
> > > > > > > > > > [3] Park, Seong-Jin, et al. "Meta variance transfer: Learning to augment from the others." International Conference on Machine Learning. PMLR, 2020. [paper link](http://proceedings.mlr.press/v119/park20b/park20b.pdf).

---

### Official Review · Reviewer_ijae · 2021-11-03

**Correctness:** 3
**Technical Novelty And Significance:** 3
**Empirical Novelty And Significance:** 3
**Recommendation:** 6
**Confidence:** 4

**Main Review:**

Strength
1. This paper proposes latent augmentation (LA), a new augmentation method to diversify training samples in the latent space. By sampling variations learned from the base classes and adding to novel class samples, LA can essentially generate more diversified samples and improve the model generalization ability.
2. Extensive experiments show the proposed LA improves the baseline counterparts on the three low-shot learning histology image classification tasks, although the improvements on the out-domain task are much smaller.
3. Ablation experiments provide more details about the influence of prototype number and augmentation times in latent augmentation.

Weakness
1. The proposed latent augmentation relies on the random initialization of k-means clustering. How does the randomness affect the result? In other words, how should the seed for K-means clustering be chosen? If a different seed is used, will the performance be much worse?
2. The variation from base classes can also be calculated from the whole dataset without K-means clustering. It seems to be a simple alternative to be compared.
3. The explanation of why CLP models generalize better than FSP models is not convincing. While the lack of dominant object in histology images is a major difference compared to ImageNet natural images, how does it result in the higher global-local feature similarity in higher feature level? Also, what is absolute and relative similarity in Figure 4?
4. Why are no regularization techniques used during fully supervised pre-training?
5. In page 7, section "DA vs. LA, and number of augmentation times", is the computation budget of LA lower than "DA", instead of "LA" again?

**Summary Of The Paper:**

This paper studies the low-shot learning problem with application to histology image classification. Specifically, contrastive learning pre-training is used for representation learning in an unsupervised manner. For low-shot classification, a new latent augmentation is proposed to augment samples by transferring variations from base classes to novel classes in the latent feature space. Three low-shot classification tasks are set up with different levels of domain shift, i.e., near-domain, mixture domain, and out-domain, using public histology image datasets. Experiments are conducted to verify the proposed method's effectiveness for these three low-shot learning tasks.

**Summary Of The Review:**

This paper proposed a new augmentation method for low-shot classification in histology images. It is well-motivated and reasonable. The experiment results also confirmed its effectiveness. There are also some issues regarding the proposed method, i.e., the influence of K-means clustering randomness, and the explanation of CLP's superiority over FSP. Overall, this paper's quality is good.

---

> ### Author Response · Authors · 2021-11-19
> **Response to Reviewer ijae (1/2)**
>
>
> We thank Reviewer ijae for the detailed comments and the positive feedback on our efforts.
>
> Below we address each concern separately.
>
> > Q1. How does the randomness of KMeans affect the results?
>
> **A1:** We use a fixed random seed for reproducibility during our experiments. Now, we add more experiments during the rebuttal period to confirm LA's robustness.
>
> The table below reports the results of the same ablation task specified in our paper (300 meta-tasks under the 5-shot setting) using different random seeds. Their numerical differences are within a 95% confidence interval, i.e., LA performs stably under different random seeds. We have revised our paper accordingly to demonstrate this point in Section 4.3 **Additional ablation studies** and Appendix D.4.
>
>
> | Random Seed   | Base       | Novel  | HMean|
> |:--------------:|:----------:|:----------:|:----------:|
> | Baseline      | 85.85±0.78 | 53.27±1.63 | 65.74±1.06 |
> | 10            | 86.85±0.84 | 65.70±1.40 | 74.81±1.05 |
> | 20            | 87.34±0.80 | 66.74±1.33 | 75.66±1.00 |
> | 30            | 87.71±0.81 | 66.48±1.35 | 75.63±1.01 |
> | 40            | 87.31±0.80 | 65.04±1.39 | 74.55±1.02 |
> | 50            | 87.83±0.78 | 66.40±1.35 | 75.63±0.99 |
> | 66 (default)  | 87.37±0.81 | 66.51±1.39 | 75.53±1.03 |
> | Average over 6 seeds | 87.40±0.81 | 66.14±1.37 | 75.30±1.01|
>
>
> > Q2. The variation from base classes can also be calculated from the whole dataset without K-means clustering. It seems to be a simple alternative to be compared.
>
> **A2:** Thanks for pointing out this valuable point, which is also suggested by the Reviewer hJxe.
>
> Following the suggestion, we conduct an additional ablation study to explore more types of covariances. We use notations of GMM in scikit-learn library [1], and define more covariance types:
> 1) ``Tied``, where all clusters share a covariance matrix estimated from the entire base dataset.
> 2) ``Diag``, where each cluster has its own diagonal covariance matrix, i.e. diagonal elements are a variance vector and non-diagonal elements are zeros
> 3) ``Spherical``, where each cluster has its own single scalar-variance shared by all feature dimensions.
>
> Our default setting is "Full". The table below reports the results in the ablation task.
>
> | Cov Type      | Base       | Novel  | HMean|
> |:--------------:|:----------:|:----------:|:----------:|
> | ``Baseline``  | 85.85±0.78 | 53.27±1.63 | 65.74±1.06 |
> | ``Tied``          | 79.35±1.08 | *65.32±1.21* | 71.65±1.14 |
> | ``Diag``          | *85.91±0.88* | 62.66±1.42 | *72.46±1.08*|
> | ``Spherical``     | 85.78±0.87 | 62.00±1.39 | 71.97±1.07 |
> | ``Full (default)``| **87.37±0.81** | **66.51±1.39** | **75.11±1.01** |
>
> LA improves baseline with all types of covariance with respect to Harmonic mean metrics. Using the covariance estimated from the entire dataset, i.e. ``Tied``, however, hurts base class performance.  Our default setting (``Full``) achieves the best performance.

---

> > ### Author Response · Authors · 2021-11-19
> > **Response to Reviewer ijae (2/2)**
> >
> > > Q3.1. What is absolute and relative similarity in Figure 4? (We answer the latter question ahead.)
> >
> > **A3.1:** We measure the cosine similarity here. Let $F^{H\times W \times C}$ be a feature map, and $F_{ij}$ be the feature vector at position $i,j$.
> >
> > The absolute similarity shows the similarity between the global average pooled representation $z=\frac{\sum_{ij} F_{ij}}{HW}$ and each local features $F_{ij}$. It is computed by $AbsSim(z, F_{ij}) = \frac{z\cdot F_{ij}}{||z|| ||F_{ij}||}$.
> >
> > The relative similarity is the min-max normalized absolute similarity over all feature vectors inside an image, i.e. $RelSim(z, F_{ij}) = \frac{AbsSim(z, F_{ij}) - \min_{ij}(AbsSim(z, F_{ij}))}{\max_{ij}(AbsSim(z, F_{ij})) - \min_{ij}(AbsSim(z, F_{ij}))}$.
> >
> > Initially, our purpose was to see how features "compete" with each other in CLP and FSP by visualizing the relative similarity.
> > We later found that CLP models have higher global-local similarity both on seen base classes and unseen novel classes. In contrast, FSP models tend to focus on the most discriminative parts.
> >
> > We have revised the caption of Figure 4 to clarify what is the absolute and the relative similarity.
> >
> > > Q3.2. The explanation of why CLP models generalize better than FSP models is not convincing. While the lack of dominant object in histology images is a major difference compared to ImageNet natural images, how does it result in the higher global-local feature similarity in higher feature level?
> >
> > **A3.2:** We sincerely appreciate this comment. It triggers our deeper analysis and further thinking of why CLP generalizes better in our problem.
> > We now modify and add more discussion on this topic in the revised paper in Section 4.4. (**Why do CLP models generalize better than FSP ones in histology images?**).
> >
> >
> > Here, we also briefly discuss our conjecture about why CLP results in a higher global-local feature similarity. In CLP, a model is required to align one view to another view of the same image [2], i.e. maximizing the similarity between two augmented views of the same image generated by random data augmentations. The augmentations include random resized cropping. Histology images usually have repeated patterns and textures. Random crops within a single histology image are likely to share the same semantics of the whole image. Thus, without knowing what part is more discriminative for this class, a CLP model may learn to produce a consistent representation for all random crops of the same image, resulting in a higher global-local feature similarity.
> >
> > However, given supervision, an FSP model may learn to pay attention to the most discriminative parts. It thus generates global representations that correspond more to those parts, showing higher global-local similarity only there. For example, in the bottom row of Figure 4 in our revised paper, an FSP model mostly pays attention to the "cytoplasm", one of the discriminative histology structures of colon cells. This behavior is also observed in Figure D.5.
> >
> > We provide more empirical explanations on why CLP generalizes better in our problem from the clustering visualization in the revised section 4.4.
> >
> > Again, we are thankful for what this comment induces.
> >
> > > Q4. Why are no regularization techniques used during fully supervised pre-training?
> >
> > **A4:** One of our primary goals is to study the generalization gap between FSP and CLP in our problem. We would like to show the large generalization gap in our experiment is larger than that in the ImageNet dataset. Therefore, we directly follow the fully supervised pre-training setting in OpenSelfSup [3], a self-supervised-learning-oriented codebase, to conduct our experiments. In their setting, only crops, horizontal flips, and normalization are used. We hope this would demonstrate the generalization gap is mainly attributed to the datasets difference or the difference between natural images and histological images.
> >
> > > Q5. In page 7, section "DA vs. LA, and the number of augmentation times", is the computation budget of LA lower than "DA", instead of "LA" again?
> >
> > **A5:** Thanks for the correction! We have revised this typo accordingly.
> >
> >
> > --------------------
> > ### Reference
> > [1] sklearn.mixture.GaussianMixture https://scikit-learn.org/stable/modules/generated/sklearn.mixture.GaussianMixture.html
> >
> > [2] Wang, Tongzhou, and Phillip Isola. "Understanding contrastive representation learning through alignment and uniformity on the hypersphere." ICML 2020. https://arxiv.org/abs/2005.10242
> >
> > [3] OpenSelfSup, https://github.com/open-mmlab/OpenSelfSup

---

> > > ### Comment · Reviewer_ijae · 2021-11-30
> > > **Response to authors**
> > >
> > > I appreciate the authors' feedback, which addressed most of my concerns. However, I am still confused by the explanation of "why no regularization techniques are used during fully supervised pre-training". Is it to avoid better generalization caused by regularization techniques? In that case, are regularization techniques also not used during contrastive learning pre-training? Otherwise, it wouldn't be a fair comparison.

---

> > > > ### Author Response · Authors · 2021-11-30
> > > > **Quick response to Reviewer ijae's concern**
> > > >
> > > >
> > > > > Are regularization techniques also not used during contrastive learning pre-training?
> > > >
> > > > No additional complex regularization techniques (e.g., DropBlock [1], distill regularization [2]) are applied during both contrastive learning pre-training (CLP) and fully-supervised pre-training (FSP).
> > > >
> > > > Data augmentation is an implicit and simple regularization technique that we use in both CLP and FSP. However, CLP uses stronger data augmentations, since it requires strong augmentations to avoid the trivial solution (see sections 3.1 and 3.2 in SimCLR paper [3]), e.g. discriminating instances by color distribution. Additionally, Chen et. al.[3] have also explored the effects of longer training and data augmentation for supervised pre-training in their SimCLR paper[3] (Appendix B.3). They observe no significant benefit from training supervised models longer and with stronger data augmentation. Therefore, we align settings in the self-supervised learning community to minimize the additional complexity in our study.
> > > >
> > > > Another difference is the use of optimizers, which should also bring implicit regularization. CLP, following MoCo-v3 [4], uses a LARS optimizer with small weight decay (1.5e-6), while FSP, following OpenSelfSup implementation, uses an SGD optimizer with no weight decay. In our experience, changing the optimizer or altering the weight decay parameter in a small range is not likely to cause a significant performance gap.
> > > >
> > > > [1] Ghiasi, Golnaz, Tsung-Yi Lin, and Quoc V. Le. "Dropblock: A regularization method for convolutional networks." arXiv preprint arXiv:1810.12890 (2018).
> > > >
> > > > [2] Tian, Yonglong, et al. "Rethinking few-shot image classification: a good embedding is all you need?." ECCV 2020
> > > >
> > > > [3] Chen, Ting, et al. "A simple framework for contrastive learning of visual representations." ICML 2020.

---

> > > > > ### Comment · Reviewer_ijae · 2021-11-30
> > > > > **Response to authors**
> > > > >
> > > > > Thanks for the detailed explanation. I have no further questions.

---

### Public Comment · ~Jiawei_Yang1 · 2022-02-18
**Post-rebuttal changes.**

We sincerely thank all reviewers, ACs, and PCs for their efforts in reviewing our work.

Here are some post-rebuttal changes we would like to clarify:

- Section 4.3, **Using label information**. When double-checking the experiments, we find we used a wrong number of base classes in the previous ablation study. In the rebuttal stage, we use “6” as the number of base classes. However, it should be “7” (excluding 2 classes from 9 classes gives us 7 base classes). This only affects Figure. 3-(c) that now unsupervised LA with only “the number of classes” information can be comparable as or even surpass supervised DC. Previous experiments using K=6 for KMeans clustering would give slightly worse results.

- Comparisons and discussions to *delta-encoder*.We thank AC for pointing out valuable related work. We now have added comparisons and discussions to delta-encoder for ablation task in Section 4.3 (Additional ablation studies) and Appendix D.7. Our LA still outperforms delta-encoder.

---

### Decision · Program_Chairs · 2022-01-20

**Decision:**

Accept (Poster)

**Comment:**

Paper presents an approach and evaluation setting for few-shot learning in histology images. The approach leverages contrastive learning pretraining, and latent augmentation (LA) for data augmentation. The evaluation examines in-domain few-shot learning, mixed domain few-shot learning, and out of domain few-shot learning.

Latent augmentation is an approach to learn how categories vary between samples within unsupervised clusters in a base dataset, and transfer that variation to the few-shot sampled classes.

Pros:
- A couple reviewers have claimed as a strength the novelty of the proposed latent augmentation method, but as other reviewers point out, there is much work in this field, some of which wasn't cited (i.e. Delta-Encoder, NeurIPS 2018).
- The latent augmentation method is simple to implement, and outperforms standard input augmentation approaches.
- The paper is rich in content and details of experiments.
- Examining learning over a variety of domain shift settings is interesting.
- Shows contrastive learning can outperform supervised pretraining for this application domain.

Cons:
- Multiple reviewers raise concerns about technical novelty. This work applies mostly previously proposed methods, or variations thereof, to the domain of medical imaging. May be more suited to a medical imaging venue.
- Some of the results are consistent with prior reports, such as finding that self-supervised learning can outperform supervised pretraining. In that regard the results are not surprising.
- One reviewer raised issues about lack of comparison to other relevant few-shot works. Authors argue that fine-tuning is a competitive baseline. Authors did add comparison to one other variation augmentation approach, distribution calibration. But as mentioned, delta-encoder is a very related work, which has not been cited nor compared against. Biggest difference is that delta-encoder uses labels, but the unsupervised clusters can trivially be supplied as labels in this setting. AC feels authors should have done a more comprehensive comparison to related learned augmentation works.
- Authors initially did not address how latent augmentation is affected by random seeds, but authors have replied to reviewers with additional data.

Reviewer consensus, excluding 1 reviewer, favors accept, though significant concerns regarding technical novelty and comparisons to other relevant works persist (especially in regards to works that learn how to augment as the proposed LA method does).